# HOMIEBOT: AN ADAPTIVE SYSTEM FOR EMBODIED MOBILE MANIPULATION IN OPEN ENVIRONMENTS

## ABSTRACT

Embodied Mobile Manipulation in Open Environments (EMMOE) is the challenge that agents understanding user instructions and executing long-horizon everyday tasks in home environments. This challenge encompasses task planning, decision-making, navigation and manipulation, and is crucial to develop a powerful home assistant capable of autonomously completing daily tasks. However, the absence of a holistic benchmark, data incompatibility between large language models (LLMs) and mobile manipulation tasks, the lack of a comprehensive framework, and insufficient dynamic adaptation mechanisms all continue to hinder its development. To address these issues, we propose EMMOE, the first unified benchmark that simultaneously evaluates high-level planners and low-level policies, and new metrics for more diverse evaluation. Additionally, we manually collect EMMOE-100, the first everyday task dataset featuring detailed decision-making processes, Chain-of-Thought (CoT) outputs, feedback from low-level execution and a trainable data format for Large Multimodal Models (LMMs). Furthermore, we design HOMIEBOT, a sophisticated agent system which integrates LMM with Direct Preference Optimization (DPO) as the high-level planner, small navigation and manipulation models as the low-level executor. Finally, we demonstrate HOMIEBOT's performance and methods for evaluating different models and policies.

## 1 INTRODUCTION

The embodied agent is defined as the integration of artificial intelligence (AI) with a physical entity, enabling it to perceive, learn and interact with the environment just like a human. In recent years, developing agents capable of understanding human instructions and autonomously completing daily tasks has become an increasingly popular challenge (Song et al., 2023; Yenamandra et al., 2023). This challenge spans multiple fields, including AI, robotics, computer vision, natural language processing and so on. Traditional frameworks for embodied agents, such as imitation learning (IL) and reinforcement learning (RL), have shown limitations on generalization and transfer ability (Shen et al., 2021; Lightman et al., 2023; Rafailov et al., 2024). Moreover, how to enable robots to actively explore and adapt to new environments while reducing the reliance on prior knowledge remains a significant challenge.

Recently, breakthrough developments in LLMs have shown great potential in complex embodied scenarios (Driess et al., 2023; Chen et al., 2023; Wang et al., 2023). LLMs excel in various natural language tasks and demonstrate strong generalization ability. Through advanced prompting techniques such as CoT (Wei et al., 2022), the logical reasoning ability of LLMs has been further improved (Wang et al., 2022; Fu et al., 2022). Visual Language Models (VLMs) enable LLMs to process visual input and allow agents to reason and make decisions based on visual observations of the environment, thus enhancing their perception and understanding abilities. The development of LMMs has further expanded the application of embodied agents into real-world settings. Moreover, with the ongoing development of demonstrations, advanced simulators (Szot et al., 2021; Kolve et al., 2017) and diverse datasets (Das et al., 2018; Li et al., 2023a; Shridhar et al., 2020), the integration of large models and embodied agents is expected to become the next wave in AI, potentially marking a crucial breakthrough in the advancement toward physical robotics.

Although LLM-driven embodied agents have been successfully applied to many downstream tasks, they still face several significant challenges when combined with mobile manipulation tasks:

1). Many benchmarks for high-level planning focus on question-answering tasks and decision-making, while benchmarks for low-level policies vary across different skills, a more sensible evaluation approach would integrate both levels into long-horizon tasks and assess the whole success rates. Moreover, the heavy reliance on simulators for evaluation and incomplete metrics also limits further developments toward real-world. 2). Robotics data for IL or RL is always not trainable for LLMs, which require dialogue-style data. Besides, while LLM training requires a large amount of data, obtaining an equivalent volume of robotics data is highly challenging. LLM prefer to output diverse human-style instructions, whereas agents require more precise and practical instructions. The incompatibility and lack of data further complicates aligning the abilities of LLMs with the needs of embodied agents. 3). To perform human-like long-horizon tasks, an agent needs to integrate multiple abilities like task planning, decision-making, navigation and manipulation into a comprehensive framework. However, existing frameworks fail to satisfy all these requirements simultaneously. 4). The agent must also have strong adaptability to make adaptation based on feedback and react with dynamic environment.

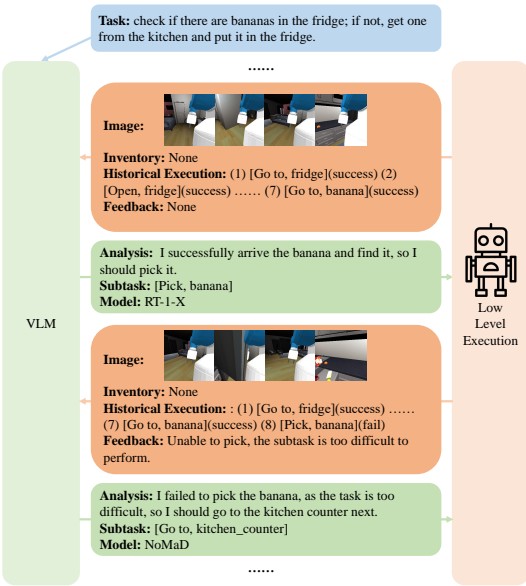

Figure 1: **An illustration of running pipeline of HomieBot.** After receiving images and feedback, LMM generates analysis, specific subtask and model choice for low level executor to perform.

To handle these issues and advance the development of embodied agents, we propose EMMOE as an open challenge, which requires agents understanding and executing long-horizon everyday tasks in home environments. Besides, we manually collect EMMOE-100, the first everyday task dataset featuring detailed task-planning processes, analyses of each output, correction during execution and a trainable data format for LMMs, which will facilitate the alignment of LMM capabilities with specific embodied tasks. Finally, we introduce HOMIEBOT, a sophisticated agent system that integrates both high-level and low-level models to complete EMMOE challenge. It is also equipped with multiple error detection and adaptation mechanisms, an execution demonstration is shown in Fig.1.

In particular, our paper makes the following contributions:

- We propose EMMOE, the first unified benchmark for both high-level and low-level embodied tasks with three novel metrics for more advanced evaluation.
- We collect EMMOE-100, the first everyday task dataset featuring CoT outputs, diverse task design, re-plan processes, SFT and DPO (Rafailov et al., 2024) dataset built on it.
- We design HOMIEBOT, a sophisticated agent system which integrates models at different levels, multiple error detection and adaptation mechanisms to complete EMMOE tasks.

## 2 EMMOE BENCHMARK

### 2.1 EMMOE-100 DATASET

Leveraging the interactive environments and robots provided by Habitat HAB 2.0 (Szot et al., 2021), we collect EMMOE-100, a dataset consists of 100 complex everyday tasks. We sample 100 different episodes from the Replica Challenge (Szot et al., 2021) to build our scenarios and design various tasks. In addition to typical tasks like *pick something and place it somewhere*, we also design other types of tasks: complex long-horizon tasks which consist of at least ten subtasks; open-ended tasks that allow multiple possible solutions; logical tasks that provide vague description and require logical reasoning; human-style tasks which are described in a natural conversation style. As these attributes

Table 1: **Dataset Comparisons** EMMOE-100 is the first dataset to integrate mobile manipulation tasks with embodied task planning, decomposing long mobile manipulation trajectories into discrete actions then executed by low-level policies in continuous space.

| | Low-level Policy Selection | Embodied Task-planning | Manipulation | Navigation | Human Annotations | Re-plan | LMM-Trainable Format | CoT Analysis | Open-ended Instructions |
|---|---|---|---|---|---|---|---|---|---|
| OVMM(Yenamandra et al., 2023) | ✗ | ✗ | Continuous | Continuous | ✗ | ✗ | ✗ | ✗ | ✗ |
| BEHAVIOR-1K(Li et al., 2023a) | ✗ | ✗ | Continuous | Continuous | ✗ | ✗ | ✗ | ✗ | ✗ |
| ALFRED(Shridhar et al., 2020) | ✗ | ✓ | Discrete | Discrete | ✓ | ✗ | ✓ | ✗ | ✗ |
| Octopus(Yang et al., 2023a) | ✗ | ✓ | Discrete | Discrete | Generated | Generated | ✓ | Generated | ✗ |
| Habitat 2.0 HAB(Szot et al., 2021) | ✗ | ✗ | Continuous | Continuous | ✗ | ✗ | ✗ | ✗ | ✗ |
| VirtualHome(Puig et al., 2018) | ✗ | ✓ | Discrete | ✗ | ✓ | ✗ | ✓ | ✗ | ✗ |
| ManiSkill-2(Gu et al., 2023) | ✗ | ✗ | Continuous | Continuous | ✗ | ✗ | ✗ | ✗ | ✗ |
| **EMMOE-100** | ✓ | ✓ | Continuous | Continuous | ✓ | Human | ✓ | Human | ✓ |

are not contradictory, a task can possess multiple attributes simultaneously, we also provide a task list and detailed task statistics in Appendix B.

Notably, EMMOE is not a subset of the Replica dataset and we only use its configuration files to construct our desired scenarios. We manually controll the robot to complete all tasks and decompose the execution process into several subtasks, finally we get 966 subtasks in total. In addition to the basic text descriptions, we also provide each subtask with first-person view images, along with detailed annotations to explain the reasoning process behind the execution. Moreover, unlike previous datasets that assume a fully successful process, we intentionally design some failed subtasks and provide their corresponding re-plan corrections to enhance the robustness of the dataset, detailed comparisons between EMMOE-100 and other mobile manipulation or embodied task planning datasets are shown in Table 1. A demonstration of our task and all its subtasks is shown in Fig.2.

## 2.2 EVALUATION METRICS

The most fundamental metrics in task planning are Success Rate (SR) and Goal-Condition Success (GC) (Shridhar et al., 2020). SR measures the rate of successful completions across all tasks, while GC is the ratio of goal conditions achieved by the end of an episode. An episode is considered successful only if GC reaches 100%. However, this evaluation metric has clear limitations when applied to our tasks. First, our task not only focuses on the final result but also pays attention to the execution process while GC only checks the final status. Additionally, though GC is effective for evaluating tasks with a single ending, it is not suitable for open-ended tasks. Human-style instructions are often open-ended and won't specify where to go or what object to get, leading to situations where the agent successfully completes the task in a different way but is still considered as a failure for not meeting pre-defined goal conditions. Moreover, setting new goal-conditions requires a deep understanding of scene information and rule-based languages like PDDL, which complicates the collection and development of new datasets. To overcome these limitations and provide a more diverse assessment of the model's capabilities, we propose the following new metrics.

**Task Progress** To better measure the task execution process and success rate, we propose a new metrics: Task Progress (TP). The calculation method for TP is as follows:

$$TP = \max_{k_i \in K_T} \left( \frac{\text{len}(k_i^{\text{check}})}{\text{len}(k_i)} \right) \tag{1}$$

$k_i$ is the $i$-th keypath in the keypath set $K_T$ for task $T$, where a keypath is defined as an ordered set of all necessary steps required to complete a task. During calculation, nodes must be checked in the exact order specified by the keypath, any node that passes evaluation will be added to another ordered set $k_i^{\text{check}}$, the ratio between the length of $k_i^{\text{check}}$ and the length of $k_i$ is used to calculate final TP. Each task is assigned with multiple keypaths, representing different possible ways to complete the task. Then the task's TP is defined as the maximum ratio across all its keypaths, the task is considered successful only when its TP reaches 100%.

The proposal of TP brings many benefits. First, it does not impact the calculation of previous metrics, though the length of different keypaths may vary, the episode length remains fixed. Besides, since keypaths are written in natural language and key node detection only requires high-level subtasks and execution status, the creation of keypaths is greatly simplified. This allows researchers to quickly

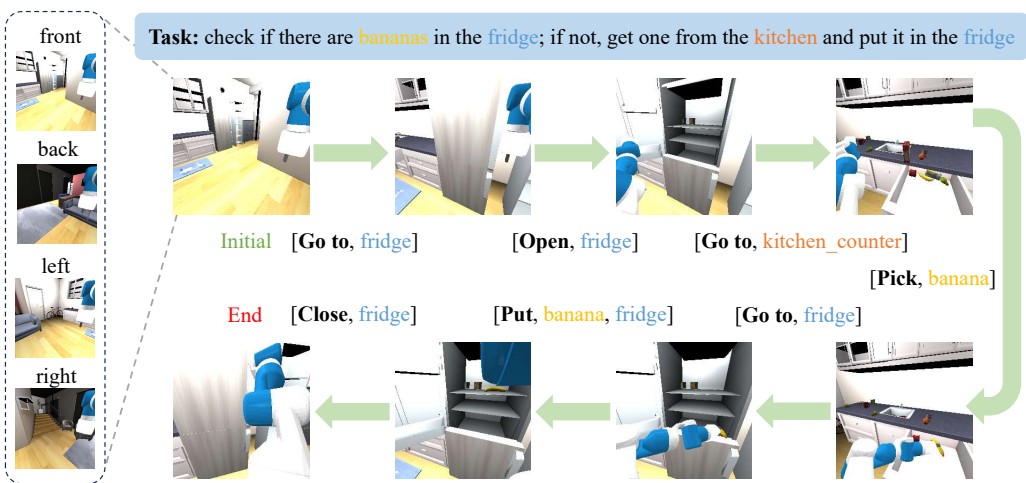

Figure 2: **Data example in EMMOE-100 dataset**. A key feature in our dataset is the emphasis on the execution process. In this task, the agent must check the fridge first; Otherwise, even if the agent finally gets a banana from the kitchen, it will not be considered as a success.

define new tasks and keypaths when collecting EMMOE datasets in new environments. Finally, this approach enables detection and evaluation in real-world settings, where writing PDDL files for evaluation is not feasible. More details about TP and keypath are in Appendix B.1.

**Success End Rate** To develop an agent that can automatically complete assigned tasks without human intervention or background information, it is crucial to equip it with the ability to judge when to stop. Without this capability, even a successfully completed task could result in an endless loop. Therefore, we propose a new metric Success End Rate (SER) to evaluate whether the agent has the ability to determine the appropriate timing for termination, and the calculation method is as follows:

$$SER = \frac{len(S)}{\sum_{t \in M} \text{count}_t(\text{end})} \tag{2}$$

where $M$ is the set of all trajectories, $S$ is the set of successful trajectories, $\text{count}_t(\text{end})$ equals 1 if the final action in the trajectory $t$ is $End$ or 0 otherwise, and $len(S)$ is the number of successful task trajectories. This metric is significant for testing fully automatic embodied agents in the future. Once it exceeds a certain threshold or even reaches 100%, metrics like GC or TP will no longer be needed to calculate the SR, as the agent would already have the ability to correctly determine when to terminate the task. Calculation demonstrations are in Appendix B.1.

**Success Re-plan Rate** In the real-world deployment of agents, the ability to quickly adapt to the environment and adjust from failure to complete tasks is crucial as the cost of failure is often unacceptable. However, there is a lack of an appropriate metric to measure this adaptive capability. Therefore, we propose Success Re-plan Rate (SRR), which measures the model's ability to effectively re-plan for task success. It is calculated as follows:

$$SRR = \frac{\sum_{t \in S} \text{count}_t(\text{replan})}{\sum_{t \in M} \text{count}_t(\text{replan})} \tag{3}$$

where $\text{count}_t(\text{replan})$ is the number of re-plans in trajectory $t$. A higher SRR value indicates powerful generalization and adaptability of the model. More demonstrations can be found in Appendix B.1.

Figure 3: **Overview of HomieBot** HomieBot leverages two modules: High-Level Planning and Low-Level Execution to complete complex daily tasks.

## 3 HOMIEBOT

### 3.1 OVERVIEW

In this section, we'll introduce how our HomieBot accomplishes EMMOE tasks and the overview of its frame is shown in the Fig.3. The process of EMMOE tasks can be described as: the agent needs to make embodied decisions based on current environments and historical execution records in open environments, then navigates and manipulates within a continuous space, the obtained results and feedback will be used for the next decision. We divide this process into two main parts: High Level Planning (HLP) and Low Level Execution (LLE). HLP is responsible for embodied decision-making and planning adaptation while LLE handles continuous execution and provide feedback to HLP. We will describe HLP in Section 3.2 and LLE in Section 3.3.

### 3.2 HIGH LEVEL PLANNING (HLP)

The key challenge in long-horizon planning is to ensure LLM-generated plans are practical, especially when static plans fail to adapt to dynamic changes. Agents need to continuously interact with the environment to refine and adjust plans based on real-time feedback. We select Video-LLaVA (Lin et al., 2023) as our base planner model $M$ and fine-tune it with our EMMOE data. Additionally, we design elaborate input and output instructions to facilitate dynamical adjustment during execution.

**Multimodal Instruction**    We decompose a long mobile manipulation trajectory into several sub-tasks, and our multimudal instruction $I$ is shown as follows:

$$I = \{o_{1\sim4}, s, T, inv, h, f\} \tag{4}$$

In the visual part, four first-person view images $o_{1\sim4}$ correspond to four directions: front, left, back and right. In the text part, system information $s$ and user task $T$ remain fixed throughout the conversation, reminding the agent of its responsibility. Feedback $f$ indicates the state of the last execution and error information if failed, and it will also be used to update other parts of $I$. Inventory $inv$ reflects what the agent currently holds, mainly to prevent generation of illogical actions, $inv$ will be updated based on both $f$ and the type of the last action. Execution history $h$ logs all previous subtasks and their results. After receiving $f$, the last subtask and its result will be updated into $h$. Notably, unlike some previous methods would first perform visual detection (Song et al., 2023) or input background information (Yang et al., 2023a) into the LLM, all background information including BEV (Bird's Eye View) images are prohibited and we directly input $o_{1\sim4}$ to LMM, which means the planner must strengthen its intrinsic capability to generate more reasonable outputs.

**Json-format Output**    Since information requirements vary from different low-level policies, a standardized output format is essential to accommodate these diverse requirements. Besides, it allows each module to function independently while simplifies module replacement and maintenance. Therefore, we define our output in the following uniform format:

$$O = M(I) = \{A, S, m\}, S = \{\texttt{action}, \texttt{target}\} \tag{5}$$

$A$ represents the analysis of each outputs, which is inspired by works like CoT (Wei et al., 2022), before generating final outputs, planner model $M$ is expected to summarize previous executions and current situations, analyze what to do next and then provide the subsequent subtask $S$. To ensure the feasibility of the output, $\texttt{action}$ in each subtask must be chosen from the available action list. Similarly, $m$ which represents the selected low-level models or policies, is also restricted to the a given model list. To fully separate decision-making from low-level execution and avoid over-reliance on the simulator, we impose fewer restrictions on $\texttt{target}$, which can be an object or a location, but it must be observable in the provided images and deemed necessary for completing the task.

### 3.3 Low level Execution (LLE)

After getting subtask $S$, model choice $m$ and inventory $inv$ from HLP, LLE would convert them into precise model-calling instructions. Error detection occurs at different stages to monitor the execution process. Once execution is completed or terminated, LLE will send environmental images and feedback to HLP for subsequent decision-making. Considering the limitations of the simulator, we set up six executable skills (see Table E1 for more details). Since different models require different input information for execution, the performance of models utilizing background information will certainly differ from those that do not. Therefore, to ensure fairness in model selection, we establish two different settings based on whether the simulator's background information is required.

**Execution With Background Information**   More specifically, execution with background information means that the selected model needs to directly obtain precise position and status information of the target from the simulator. As M3 (Gu et al., 2022) demonstrates excellent performance across all the skills we define when utilizing background information from the habitat, we choose it as the unique low-level model. In this case, $m$ is masked and the model choice is always M3. Since M3 requires specific background names and our target cannot be directly recognized by it, we apply several processing steps before passing the information to M3. This ensures that the information is converted into a granularity that M3 can recognize. More details are in Section 4.3.

After the execution is completed, in addition to the text and image data provided to HLP, LLE also captures the entire execution process in a video. At the end of each task, we will obtain a complete trajectory video with detailed annotations for each step. This means that HomieBot has the potential to bridge the gap between robot data and LLM data as the entire execution process is fully automated and the user only needs to set up the scene and input instructions. The video data can be used for IL in robotics, while the text and image data can be used for LMM training.

**Execution Without Background Information**   Without background information means that the agent can only rely on the information captured by its own sensors and its intrinsic ability to complete the task. As shown in Table E2, we set two manipulation models and two navigation models to perform different actions and model choice $m$ from HLP will determine which model to use. The manipulation models include RT-1-X (Padalkar et al., 2023) and Octo (Team et al., 2024b), RT-1-X is used for $Pick$ and $Place$, and Octo is used for $Open$ and $Close$. The navigation models consist of NoMaD (Sridhar et al., 2024) and PixNav (Cai et al., 2024). NoMaD specializes in image navigation and is suitable when the $target$ is a spot or large objects, whereas PixNav excels at pixel-level and object navigation, making it ideal when the $target$ is an object.

The primary criterion for selecting low-level models in this setting is that they should be lightweighted rather than large models, aligning closely with our motivations for this setting. We aims to prepare for the deployment and evaluation of agents in real-world settings, where background information is often lacking. Real-world operations require high real-time performance, and the inference speed of LLMs remains a challenge. By breaking down long-horizon tasks into action primitives, we can leverage task-specific small language models (SLMs) for these actions, thus reducing time costs and avoiding an overly large framework.

**Error Detection**   Since errors may occur at different stages during the execution of long-horizon tasks, to facilitate better communication with HLP and provide more detailed error information, we

design four major types and several sub-types of error detection. **Logical error** *L1*: the agent already holds an object but still attempts to pick/open/close; *L2*: the agent holds nothing but attempts to put; *L3*: the agent attempts to pick/put object in a closed container; *L4*: the agent attempts to open/close an non-interactive object. **Distance error** *D1*: the agent is too far from the target, preventing interaction with the target object; *D2*: the agent is too close to the target,the robotic arm is hindered from extending properly during interaction. **Format Error** *F1*: The output action or model is not in the available list; *F2*: The output target does not exist in the scene or can not be recognized by low level models. **Execution Error** *E1*: failure is due to limited capabilities of low-level models or policies; *E2*: improper execution may lead to the inventory information being accidentally updated. More classification and detection details are in Appendix E.

## 4 EXPERIMENTS

### 4.1 DATA AUGMENTATION

**SFT Augmentation**  Previous work(Zhang et al., 2024b) has demonstrated that a standardized data format would significantly enhance model training and evaluation. To this end, we write a uniform script (see in Appendix F.1) to convert EMMOE-100 data into fixed-format conversation data. During this process, all failed subtasks will be skipped as they are treated as junk data for the SFT dataset and we initially obtained 930 SFT data in this way, which is still insufficient for LLM training. To expand the dataset, we use GPT-4o (Achiam et al., 2023) to regenerate text descriptions of tasks and the analysis of each subtask for three times. This approach not only enhances the diversity of instructions, allowing the LLM to adapt to different user input styles, but also helps to avoid introducing additional inaccuracy or inconsistency. Finally, we obtain 3720 SFT data in total.

**DPO Augmentation**  DPO (Rafailov et al., 2024) training has a strict requirement for data format, which must include *prompt*, *chosen* and *rejected*. For the $i$th subtask and its input instruction $I_i$, if the execution of model output $O_i$ fails but the next output $O_{i+1}$ succeeds after re-plan, we will choose $I_i$ as the *prompt*, $O_i$ as the *rejected* and $O_{i+1}$ as the *chosen*. Although this approach aligns well with the concept of preference data, the proportion of re-planed data is relatively low. Therefore, we utilize following methods to construct new DPO data. **Order Change**: We shuffle the order of successful subtasks, treating output $O_i$ as *chosen* and $O_{i+1}$ as *rejected*. This approach aims to help LLMs learn the logical relationships between subtasks, particularly understanding the optimal order of actions. **Action Change**: To standardize the model's output and reduce responses outside the action list, we replace subtask actions with non-standard names or actions not in the list. **Model Change**: To enable the LLM owns the ability to select the appropriate low-level model for a given scenario, we replace the model choice with other models in the model list. Finally, we get 10104 DPO data in total. More visualized processing flows and data samples can be found in Appendix F.2.

### 4.2 MODEL TRAINING

**Training Details**  We select 90 tasks from EMMOE-100 as our training tasks. Using the methods described in Section 4.1, we obtain 3,316 SFT training data and 8,984 DPO training data. Then we choose Video-LLaVA-7B (Lin et al., 2023) as our base model and conduct a two-stage training process. In the first stage, we fine-tune the base model with a learning rate of 5e-4 on four NVIDIA A40. In the second stage, we align the fine-tuned model with DPO method and train with a learning rate of 5e-6. To prevent catastrophic forgetting and retain the model's intrinsic capability, LoRA (Hu et al., 2021) is applied in both stages, with LoRA rank set to 128 and $\alpha$ to 256 in stage one, and LoRA rank set to 8 and $\alpha$ to 8 in stage two. More training details are shown in Appendix G.

### 4.3 SETUP

**Metrics**  In addition to SR, TP, SER and SRR introduced in Section 2.2, we also choose Path Length Weighted SR (PLWSR)(Shridhar et al., 2020) as one of our evaluation metrics. PLWSR is defined as SR$\times$(length of successful trajectory) / $max$(length of expert trajectory, length of successful trajectory) and it measures the ability gap between the agent and the expert in successful trajectories.

Table 2: Performance comparison of tasks in EMMOE-100 dataset. The highest values per metric are shown in **bold**. All values are percentages.

| Model | SR | PLWSR | TP | SRR | SER |
|---|---|---|---|---|---|
| GPT-4o(Achiam et al., 2023) | 13.33 | 10.51 | 29.79 | 3.57 | 49.38 |
| Gemini-1.5-Pro(Team et al., 2024a) | 17.33 | 14.79 | 38.03 | 3.39 | 55.91 |
| Qwen2-VL-7B(Wang et al., 2024) | 1.00 | 0.50 | 16.55 | 0.59 | 25.00 |
| MiniCPM-V 2.6(Yao et al., 2024) | 0.67 | 0.57 | 14.45 | 0.06 | 40.00 |
| HomieBot-7B (SFT) | 27.67 | 20.88 | 50.27 | **9.23** | 53.90 |
| HomieBot-7B (SFT+DPO) | **30.30** | **24.66** | **51.39** | 8.72 | **60.81** |

Table 3: Performance comparison of HomieBot bewteen train and test split. The highest values per metric are shown in **bold**. All values are percentages.

| Model | Train split | | | | | Test split | | | | |
|---|---|---|---|---|---|---|---|---|---|---|
| | SR | PLWSR | TP | SRR | SER | SR | PLWSR | TP | SRR | SER |
| HomieBot (SFT) | 28.52 | 21.49 | 50.16 | 9.59 | 53.85 | **20.00** | **15.36** | **51.19** | **6.55** | 54.55 |
| HomieBot (SFT+DPO) | **31.84** | **25.82** | **52.29** | **9.69** | **60.71** | 16.67 | 14.36 | 43.39 | 3.08 | **62.50** |

**Baselines** **High Level Planner**: Modular framework and communication mechanism greatly facilitate the deployment of various LMMs into our HomieBot. We select four representative LMMs as baseline planners: GPT-4o (Achiam et al., 2023), Gemini-1.5-Pro (Team et al., 2024a), Qwen2-VL-7B (Wang et al., 2024) and MiniCPM-V 2.6 (Yao et al., 2024). GPT-4o and Gemini-1.5-Pro can be easily integrated into HomieBot after minimal adjustments to format requirements. By leveraging the in-context learning abilities and providing output examples for each inference, the other two models can also be deployed into our system. **Low Level Executor**: We extract individual skills from M3 (Gu et al., 2022) and modify their implementations. The original skills require the initial and final states of the object. We map the object name to obtain specific background information and select the nearest object. Additionally, the arm status will be reset after each execution to enhance the success rate. We also pass all environmental state information between executions to ensure environmental consistency. We provide more deployment details in Appendix H.1.

**Evaluation Dataset** All tasks in EMMOE-100 will be used for evaluation, the remaining ten tasks that are not used as training data will serve as our test set. Each task is executed three times, with a maximum step limit of 20. We will use the average result of each task for the final calculation.

### 4.4 RESULTS

We first conduct a unified evaluation since all data are unseen to baseline models, and the results are shown in Table 2. Our DPO version achieves the best performance in SR, PLWSR and TP metrics, significantly surpassing the baseline models. Additionally, it is evident that for open-source models of similar size, even state-of-the-art LMMs like Qwen2-VL-7B (Wang et al., 2024) and MiniCPM-V 2.6 (Yao et al., 2024) struggle to complete EMMOE tasks without additional training. For SER, though DPO version performs best, the improvement is not so obvious as in other metrics, Gemini-1.5-Pro even surpasses the SFT version. This is because SER reflects the model's ability to correctly judge when a task is completed and should terminate. It is not influenced by format requirements or low-level execution but is more related to the model's inherent reasoning ability. The strong reasoning capabilities of GPT-4o (Achiam et al., 2023) and Gemini-1.5-Pro (Team et al., 2024a) enable them to effectively determine when a trajectory should end without training.

However, SFT version performs best rather than DPO version for SRR. Since SRR reflects the model's ability to adapt to environments and adjust from failure, we think this could be attributed to limitations of the DPO method (Xu et al., 2024). While DPO brings unparalleled advantages in training efficiency and speed, it may compromise the model's generalization and transferability. Therefore, we evaluate HomieBot separately on training and test set, and the results are shown in Table 3. As we can observe, while DPO version performs best across all metrics in the training split, it only outperforms SFT version in SER during the test split. Besides, SRR shows a significant

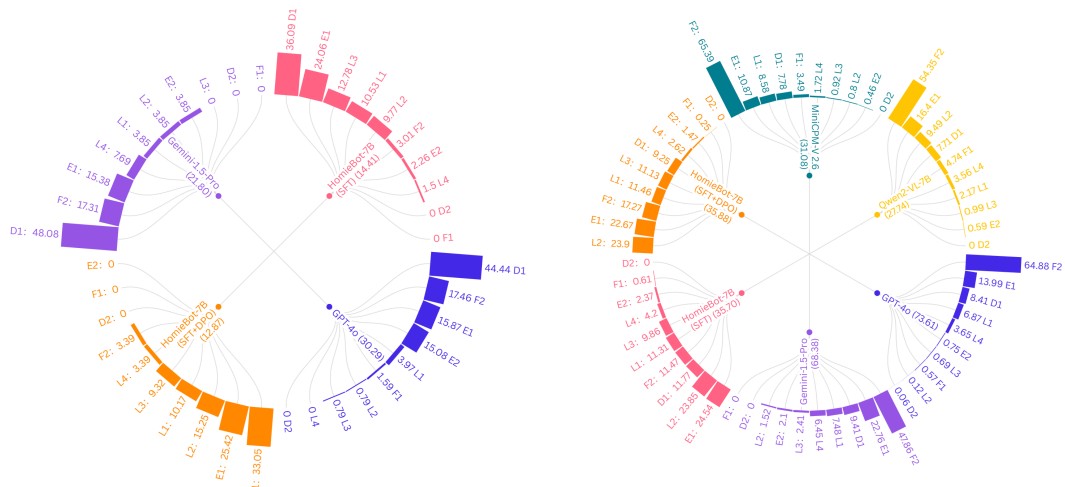

Figure 4: **Error Statistics** The left and right figures respectively depict the proportion of each error among all errors in successful and failed trajectories. Additionally, we indicate the percentage of failed execution times out of total execution times next to each model's name. Due to too few successful trajectories for Qwen2-VL and MiniCPM-V 2.6, their results will not be shown in the left figure. All values are percentages and the raw statistical data is available in Appendix H.2.

decline and even performs slightly worse than GPT-4o and Gemini-1.5-Pro. This phenomenon further confirms that the DPO method introduces certain generalization issues. Notably, SER for both versions remains stable across the training and test splits, which further demonstrates that SER is more related to the model's inherent judgment ability, and our specialized handling of $End$ during dataset construction has enhanced this ability (See in Appendix F.2).

## 4.5 ANALYSIS

To further explore the reasons for the overall low success rate and demonstrate how HomieBot can be used to simultaneously evaluate both HLP and LLE, we conduct a detailed analysis based on the results in Section 4.4. Using the error classification in Section 3.3 and recorded feedback, we collect all errors that occurred during experiments. To determine which errors are acceptable and solvable and which are the main causes of failure, we further classify them according to whether they appear in successful or failed trajectories, the results are shown in the Figure 4.

**Error Analysis** Except for $E1$ and $E2$ errors which come from low-level models, each type of error corresponds to different capabilities of LMMs. In failed trajectories, the predominant error type across all baseline models is $F2$ error, which largely exceeds other types. This suggests that the primary factors impeding baseline model's performance are physical grounding failures and model hallucinations. In practical execution cases, we observe that even models are told that objects do not exist or cannot be recognized, they may still output wrong objects or repeat mistakes after several steps. This issue has been significantly improved in our models, which also highlights the significance of LMM-trainable format data. With a small amount of data combined with our SFT and DPO data augmentation methods, LMM can build up a general understanding of the current environment, enabling outputs to be grounded and compatible with low-level models. Besides, the ratio of failed executions to total executions for two open-source models is relatively low, indicating most subtasks are successfully executed, which seems to conflict with the very poor SR. In our observations, since EMMOE involves very complex and long-horizon tasks, the execution steps can easily become very long, when the model's understanding ability is insufficient, it may fail to fully grasp previous execution content, ultimately resulting in the meaningless output. Although these subtasks can be successfully executed, they will contribute nothing to the overall task, and even worse, they will consume remaining steps and fasten task termination. In successful trajectories, the most common error is $D1$ error. This indicates that even when the model's spatial perception ability

Table 4: P represents the percentage of individual action errors relative to the total execution errors. SR here represents an average value as each skill is attempted up to three times per execution. All values are percentages and the raw statistical data is available in Appendix H.2.

| Metrics | Go to | Pick | Place | Open | Close |
|---------|-------|------|-------|------|-------|
| P | 38.49 | 49.77 | 7.30 | 3.32 | 1.11 |
| SR | 45.32 | 22.45 | 40.97 | 43.13 | 36.45 |

Table 5: The performance of each type of task is presented in the format SR (PLWSR). The highest value for each model is highlighted in red. All values are percentages.

| Model | typical | long-horizon | open-ended | logical | human-style |
|-------|---------|--------------|------------|---------|-------------|
| HomieBot-7B (SFT) | 43.75 (32.31) | 24.60 (18.70) | 18.52 (11.93) | 34.01 (25.45) | 25.24 (18.70) |
| HomieBot-7B (SFT+DPO) | 41.67 (34.24) | 28.11 (22.82) | 15.38 (11.57) | 35.86 (28.05) | 27.88 (21.78) |

is insufficient, it can be adjusted through feedback information. Typically, after a $D1$ error occurs, the model will output $Go\ to$ action based on the feedback, effectively resolving this error. We conduct more detailed case study in Appendix I.

**LLE Evaluation** Comprehensive error types allow us to evaluate HLP and LLE separately. We further classify $E1$ and $E2$ errors based on action types and count total occurrences of each action, the calculation results are shown in Table 4. It is evident that $Pick$ action has a significantly lower success rate and the highest proportion of execution errors compared to other actions.

**Task Performance** We also evaluate SR and PLWSR for each type of task defined in Section 2.1. As shown in Table 5, typical tasks are relatively easy due to straightforward processes and fewer overall steps. The most challenging are open-ended tasks, which usually have a very long total step count, with flexible processes and results, demanding powerful capabilities from both HLP and LLE models.

## 5 LIMITATIONS AND FUTURE WORKS

**Limitations** Firstly, the range of available actions and interactive objects is limited in Habitat, restricting the scope of our task design. Secondly, When the scenario is expanded to multiple rooms, the demands on navigation and memory functions will significantly increase, potentially necessitating the integration of additional memory mechanisms. Although standardized output enables uniform evaluation of different models, it will sacrifice certain information precision to meet each model's requirements, and the increasing number of model inferences will lead to additional time costs. Therefore, designing a more efficient workflow and output format is necessary. Finally, disparities among simulators and between simulators and the real world pose significant challenges for low-level model transfer and generalization, a more general and universal evaluation platform is needed.

**Future Works** Recently, more powerful simulators like Robocasa (Nasiriany et al., 2024) enable the design and collection of a wider range of everyday tasks. Besides, how to efficiently utilize historical images and videos rather than text only to improve decision-making still needs to be explored. Additionally, real-world deployment of Homie is possible since the data collection and evaluation metrics are independent. Furthermore, exploring how Homie can interact or collaborate with family members to complete daily tasks is also worthful and necessary.

## 6 CONCLUSION

In our work, we first introduce EMMOE, the first unified benchmark designed to evaluate both high-level planners and low-level policies. Then we present the collection and features of EMMOE-100 and propose three novel metrics to complement existing metrics. Next, we introduce our HomieBot and illustrate how its two main components HLP and LLE function. In experimental parts, we demonstrate how to use original EMMOE data to construct LLM-trainable SFT and DPO datasets and evaluate different models, we also conduct deep analysis based on detailed error information.

## ETHICS STATEMENT

This research utilizes publicly accessible models and simulators, ensuring that all data complies with privacy regulations and has been anonymized where required. We are aware of potential biases that may emerge in automated data generation, especially those related to gender, race, or other attributes. To address this, we have implemented measures to assess and reduce such biases and are dedicated to continuous improvement in this area. Moreover, we acknowledge the risks of misuse, such as generating misleading data, and have incorporated safeguards to prevent such uses. Our objective is to foster the responsible development and application of embodied agent technology to advance accessibility and automation, while upholding ethical standards in AI development. To support reproducibility and further research, all code and models will be openly shared.

## REPRODUCIBILITY STATEMENT

We have made several efforts to ensure the reproducibility of our work. All the training procedures, and hyperparameter settings, are described in Appendix G. Detailed information about our datasets and demonstrations can be found in Appendix B. More examples of metric calculation can be found in B.1. We provide more details and code clips of our pipeline in Appendix D and Appendix E. Codes and demonstrations of data augmentation are in Appendix F. We supply more details about the experiments and running pipelines in Appendix H For theoretical results, we provide a clear outline of the assumptions and complete proofs in Appendix I. We have also outlined any hardware and software configurations used for our experiments to further support reproducibility.

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

# Supplementary Material

The supplementary material is structured as follows:

- Related Work in Section A.
- Dataset in Section B.
- Metric Calculation in Section B.1.
- High Level Planning in Section D.
- Low Level Execution in Section E.
- Data Augmentation in Section F.
- Training Details in Section G.
- Experimental Details in Section H.
- Case Study in Section I.

## A    RELATED WORK

### A.1    EMBODIED TASKS AND BENCHMARKS

As embodied agents and LLMs develop rapidly, many embodied tasks and benchmarks have emerged. In Embodied Question Answering (EQA) tasks, EQA-v1 (Das et al., 2018), MT-EQA (Yu et al., 2019), MP3D-EQA (Wijmans et al., 2019), IQUAD V1 (Gordon et al., 2018), OpenEQA (Majumdar et al., 2024), HM-EQA (Ren et al., 2024), S-EQA (Dorbala et al., 2024) contains a variety of task range to evaluate logical reasoning abilities of LLMs. In Vision-and-Language Navigation (VLN) tasks, R2R (Anderson et al., 2018), R4R (Jain et al., 2019) and VLN-CE (Krantz et al., 2020), SOON (Zhu et al., 2021) evaluate LLM's capabilities under different navigation settings. ALFRED (Shridhar et al., 2020) Behavior series (Srivastava et al., 2022; Li et al., 2023a) focus on interactive household tasks OVMM (Yenamandra et al., 2023) involves picking and placing any object in unseen environments. Common grasping datasets include MT-Opt (Kalashnikov et al., 2021), VIMA (Jiang et al., 2022), ManiSkill2 (Gu et al., 2023), Calvin (Mees et al., 2022), BridgeData-v2 (Walke et al., 2023), RH20T (Fang et al., 2023) and Open-X (O'Neill et al., 2024). In mobile manipulation, RT series (Brohan et al., 2022; Zitkovich et al., 2023) and Mobile ALOHA (Fu et al., 2024) exhibit strong capabilities. Despite numerous benchmarks, a unified benchmark and relevant task is still missing. Traditional mobile manipulation uses IL to learn entire trajectories, complicating the evaluation of intermediate processes. In our work, we propose EMMOE, a holistic benchmark designed to assess both final results and the execution process.

### A.2    LLM-DRIVEN EMBODIED AGENTS

LLM-driven embodied agents represent cutting-edge advancements in robotics. SayCan (Ahn et al., 2022), Palm-E (Driess et al., 2023), LLM-Planner (Song et al., 2023) and EmbodiedGPT (Mu et al., 2024) combine LLMs with complex embodied tasks. TAPA (Wu et al., 2023) and SayPlan (Rana et al., 2023) use visual modules for multi-room settings. Voyager (Wang et al., 2023), STEVE (Zhao et al., 2023b), Smallville (Park et al., 2023) and Octopus (Yang et al., 2023a) use LLMs to choose pre-defined functions. L3MVN (Yu et al., 2023), ESC (Zhou et al., 2023), SayNav (Rajvanshi et al., 2023) and VLFM(Yokoyama et al., 2024) build frontier or semantic maps to navigate. ViNT (Shah et al., 2023) and NoMaD (Sridhar et al., 2024) focus on image navigation, PixNav (Cai et al., 2024) uses LLM to select target image pixel. GOAT (Chang et al., 2023) is a comprehensive navigation system. RT-2 (Zitkovich et al., 2023) is the first Visual Language Action (VLA) model. RoboFlamingo (Li et al., 2023b) and OpenVLA (Kim et al., 2024) are open-source VLA models. Leo (Huang et al., 2024) focuses on multiple QA problems. Octo (Team et al., 2024b) is a light model for arm control. ALOHA (Zhao et al., 2023a) improves action prediction through action chunking. RoboAgent (Bharadhwaj et al., 2024) enhances object detection and generalization, and LCB (Shentu et al., 2024) uses LLMs to generate implicit strategy goals. ManipLLM (Li et al., 2024) and VoxPoser (Huang et al., 2023) combine environmental perception and task execution.

## A.3 ADAPTATION FROM FEEDBACK

Discrepancies between LLM's outputs and real-world conditions always lead to execution failures. LLM-Planner (Song et al., 2023) make a straightforward re-plan. Self-Refine (Madaan et al., 2024) use single LLM as generator and evaluator. Reflexion (Shinn et al., 2024) treats LLM as the Actor and the evaluator as the Critic. ViLA (Lin et al., 2024) utilizes GPT-4V (Yang et al., 2023b) to obtain visual feedback. However, self-improvement relies heavily on prompt design and intrinsic capabilities of LLMs. If errors unrelated to planning occur, LLMs may struggle to self-correct. Inner Monologue (Huang et al., 2022) and RoCo (Mandi et al., 2024) utilizes external collision detection and feedback. DoReMi (Xie et al., 2024) sets pre-defined constrains. Nevertheless, LLMs may make same mistakes in similar situations as the model weights are not changed. SayCan (Ahn et al., 2022) trains a value function to consider both generated actions and their values. Remember (Zhang et al., 2024a) builds a memory module and retrieves similar state-action pairs. Retroformer (Yao et al., 2023) learns a retrospective model via policy gradient optimization. While RL-based adaptation mechanisms are able to adjust actions before execution, defining and training an effective value function or reward model is highly challenging. The recently popular DPO (Rafailov et al., 2024) algorithm greatly simplifies this process by requiring only a preference dataset. In our HOMIEBOT, we use DPO for model alignment, CoT (Wei et al., 2022) and self-reflection for decision-making. Additionally, error detection and feedback mechanisms are applied during low-level execution.

## B DATASET

### B.1 DATASET DETAILS

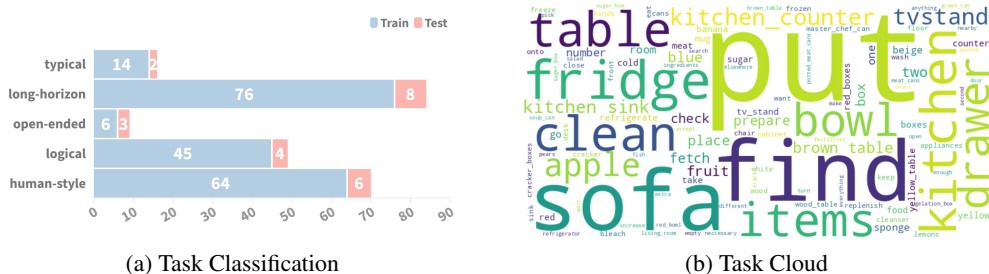

(a) Task Classification        (b) Task Cloud

Figure B1: Dataset Statistics

In terms of task classification, the long-horizon task is the most, with 84 (76 in the train set and 8 in the test set). The least task is the most difficult open-ended task, with 6 in the train set and 3 in the test set. The distribution of the five types of tasks in the train and test sets is also approximately same. In the word cloud map, we can see that put, find, sofa, etc. are popular words in our task.

### B.2 DATASET DEMONSTRATION

Here we provide a demonstration of EMMOE data, its viusal information is shown in Fig. B2, the collection method is as Section 2.1.

```
Task: check if there are anything in the kitchen drawer, if it's empty put a short can into it

Subtask1: [Go to, blue_sofa] (PixNav)
Analysis: According to my final task, I must go to kitchen first, but I can't find kitchen in my sight
and I find a blue sofa, so I should go to the blue sofa first and find kitchen later
Feedback: None
Inventory: None

Subtask2: [Go to, living_room] (NoMaD)
Analysis: I successfully arrived sofa but I still can't find kitchen, as I see living room, I should go
 to living room then keep searching kitchen
Feedback: None
Inventory: None
```

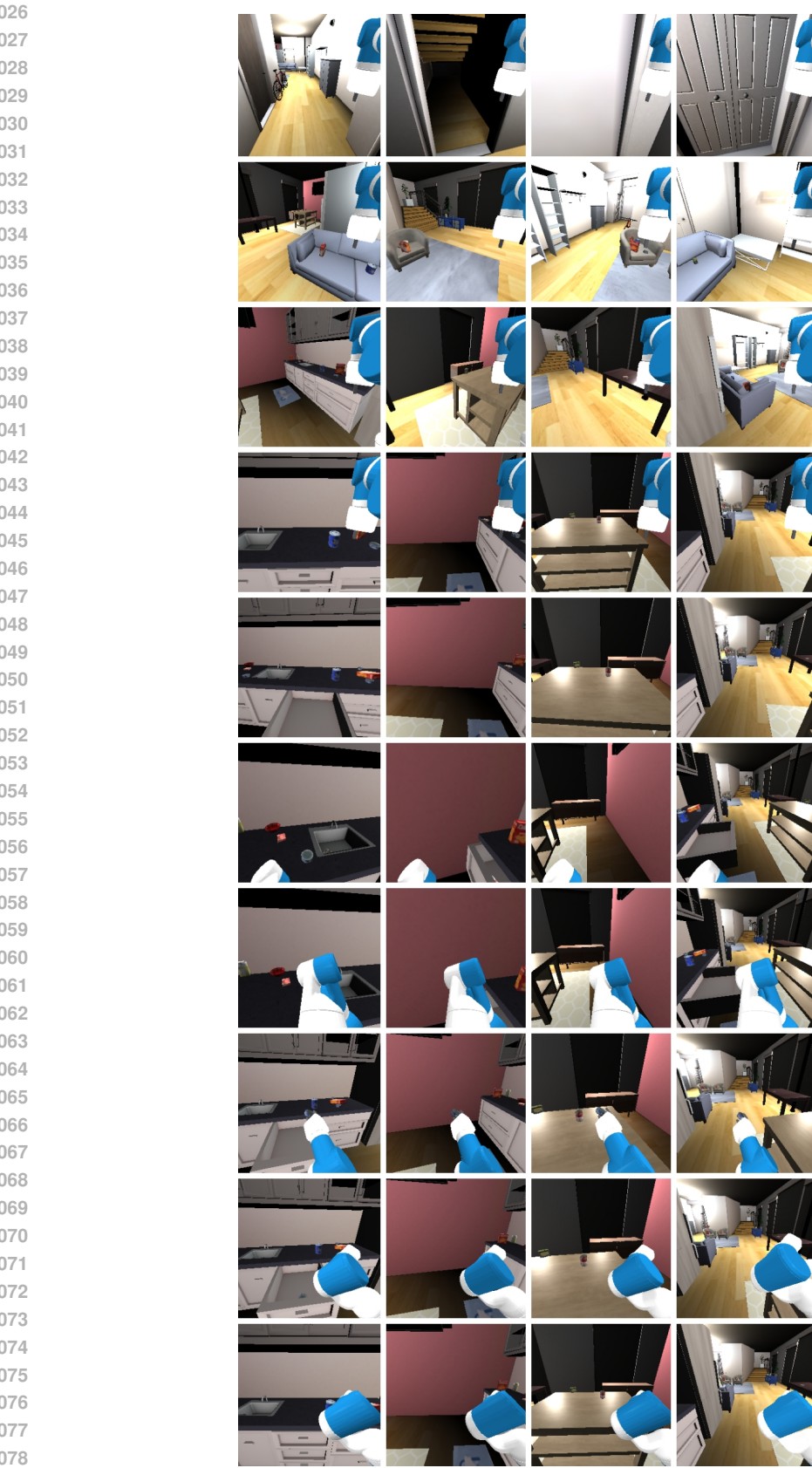

Figure B2: **Task Demonstration.** line: subtask sequence; row: front, left, back, right

```
Subtask3: [Go to, kitchen_drawer] (NoMaD)
Analysis: I successfully arrived the living room and find the kitchen drawer, so I should go to kitchen
 drawer next
Feedback: None
Inventory: None

Subtask4: [Open, kitchen_drawer] (octo)
Analysis: I successfully arrived the kitchen drawer but it's closed. To check items in the drawer, I
should open it first
Feedback: None
Inventory: None

Subtask5: [Go to, kitchen_counter] (NoMaD)
Analysis: I successfully opened the drawer and find nothing in it, according to my task, I should put a
 short can into it in this case, as I find one on the kitchen counter, I should go there next
Feedback: None
Inventory: None

Subtask6: [Pick, short_can] (RT-1-X)
Analysis: I successfully arrived the kitchen counter and find the can, so I should pick it next
Feedback: None
Inventory: short_can

Subtask7: [Go to, kitchen_drawer] (NoMaD)
Analysis: I successfully picked the short can and I should come back to kitchen drawer to put it in
Feedback: None
Inventory: short_can

Subtask8: [Put, short_can, kitchen_drawer] (RT-1-X)
Analysis: I successfully arrived the kitchen drawer and I should put the can into it next
Feedback: None
Inventory: None

Subtask9: [Close, kitchen_drawer] (octo)
Analysis: I successfully put the can into the drawer, and it's better to close the drawer next
Feedback: None
Inventory: None

Subtask10: [End]
Analysis: According to the historical execution and final task, I have finally finished the task and it
's time to end
Feedback: None
Inventory: None
```

We also provide all designed tasks here, the task design principles focus on reflecting human's real-life with a variety of common demands and task descriptions.

```
(1) fetch a frozen meat can and put it on the kitchen counter
(2) clean up the brown table and place all items in the fridge
(3) find a cold apple and put it on the kitchen_counter
(4) find an bowl and put it on the sofa
(5) find an master_chef_can on the wood_table and put it into the drawer
(6) go to the floor 2
(7) prepare neccessary ingredients to make a fruit salad and put them on the yellow_table
(8) keep the number of red_boxes on the yellow_table to 5
(9) search a blue can for me
(10) fetch one crack box and one sugar box and put them on the beige table
(11) find two cracker boxes in the room and put them on the kitchen counter
(12) check if there are apples in the fridge and put one into it if not
(13) pick all fruit on the brown table and put them on the sofa
(14) put the bowl into the kitchen cabinet
(15) find a bleach cleanser and a sponge then place them on the brown table
(16) fetch two apples from the kitchen counter and put them into the fridge
(17) clean the wood table and put all items except mug to the sofa
(18) I want to eat at the brown table and prepare a fish can for me
(19) fetch two cracker_boxes from the kitchen sink and refrigerate them
(20) check and close all kitchen facilities
(21) prepare two bowls on the brown table
(22) fetch two meat_cans from the kitchen and put them on the beige table
(23) find a mug and put it on the tvstand
(24) go to kitchen then put the red box into the drawer and put the red can into the fridge
(25) find an apple and place it on the tv_stand
(26) clean the tvstand and put all items to the sofa
(27) clean up the tv_stand and put all items in the kitchen drawer
(28) put the sponge and bleach cleanser on the sofa into the kitchen drawer
(29) freeze a sugar_box
(30) put the blue can on the kitchen_counter to the fridge
(31) find two potted_meat_cans and put them on the sofa
(32) clean up the blue table and put all items to the white cabinet
(33) find an apple and put it on the sofa
(34) take a bowl and a meat can from the kitchen and put them on the brown table
(35) clean up the kitchen sink and put fruit to the fridge other items to the kitchen_counter
(36) replenish the number of blue cans on the table to 3
(37) find two bowls in the room and put them in the kitchen sink
(38) put all cracker_boxes on the tvstand to the sofa
(39) take a yellow box and put it into the fridge.
(40) put the apple on the blue table to the sofa
(41) fetch 3 different kinds of fruit and put them on the beige table
(42) I want to eat at the brown table and prepare some fruit for me
(43) put the frozen sponge into the kitchen drawer
```

```
(44) put all bowls on the sofa to the kitchen sink
(45) get a can in the fridge and put it on the table
(46) prepare a washed apple then put it on the yellow table
(47) clean up the tvstand
(48) clean up the chair
(49) put everything in the kitchen sink onto the kitchen_counter
(50) wash the bowl on the kitchen_counter
(51) fetch two sugar boxes in the fridge and put them on the brown table, if there aren't enough sugar
boxes in the fridge, find them elsewhere in the room
(52) Prepare a soup_can and a red_bowl on the kitchen_counter
(53) put all the fruit on the kitchen_counter into the sink
(54) put the bowl on the wood_table and the apple on the kitchen_counter to the kitchen sink
(55) refrigerate all master_chef_cans on the tvstand
(56) clean up the blue sofa
(57) find a gelation_box and put it in the drawer
(58) put the cracker box in the kitchen sink to the sofa
(59) check if there is food on the sofa then put them in the fridge if so
(60) refrigerate all lemons in the kitchen drawer
(61) put all food on the sofa into the drawer
(62) take the bowl on the table to the kitchen
(63) clean up the tv_stand and place items on the kitchen_counter
(64) check if there are bananas in the fridge; if not, get one from the kitchen and put it in the
fridge
(65) fetch a yellow box from the refrigerator and place it on the table, if there isn't one, get it
from the kitchen
(66) clean the sofa and put all items on the table in front of it
(67) find an apple and place it in the drawer
(68) Put the red bowl on the blue table in the fridge.
(69) go to the second floor
(70) keep the number of red_boxes on the yellow_table to 3 and put extra red_boxes to the
kitchen_counter
(71) clean up the beige table and put all items to kitchen
(72) put all fruit in the living_room to the fridge
(73) find an apple and place it in the fridge
(74) find a bowl and a mug then put them into the kitchen sink
(75) replenish the number of pears in the fridge to 3
(76) find an apple and put it on the brown table
(77) put all lemons and apples on the sofa to the tvstand
(78) put all bowls in the open drawer onto the kitchen_counter
(79) clean up the sofa and put all items into the drawer
(80) clean up the sofa and place all items on the nearby chair
(81) freeze the meat can on the blue desk
(82) check and close all appliances in the room
(83) get a cold apple and put it on the wood table
(84) check if there are anything in the kitchen drawer, if it's empty put a short can into it
(85) turn off all appliances in the room then go the door and wait
(86) prepare some food and put it on the brown table
(87) check items in the fridge then increase the number of blue cans to 2
(88) find a box and put it on the tvstand
(89) clean the table in front of you and put all items into the sink
(90) find two bananas on the tvstand and put them to the kitchen sink
(91) find the bowl in the drawer and put it to the kitchen sink
(92) get a cold fruit and prepare to wash it
(93) clean the sofa
(94) put all items on the sofa to the tvstand
(95) put all items on the blue sofa to the white desk
(96) find the sponge and put it into the drawer
(97) find two kinds of fruit and put them on the tvstand
(98) find a banana and place it in a bowl
(99) put the bowl on the brown table into the kitchen sink and put the suger_box on the tvstand to the
sofa
(100) put the green_can on the brown_table to the fridge
```

# C   METRIC CALCULATION

## C.1   TASK PROGRESS

In the task demonstrated in Appendix B, it's easy to find that to complete the task, we have to open
the drawer to see if there is anything, and then we have to finish a put operation (put short can in the
drawer). In addition to these two, we can also add some operation like, go to the drawer, close the
cook and other actions which do not influence the final success. So we get the keypath as shown
below,

```
[
    [
        "[Open, kitchen_drawer]",
        "[Put, short_can, kitchen_drawer]",
        "[End]"
    ],
    [
        "[Open, kitchen_drawer]",
        "[Put, short_can, kitchen_drawer]",
```

```
            "[Close, drawer]",
            "[End]"
        ],
        [
            "[Go to, drawer]",
            "[Open, kitchen_drawer]",
            "[Put, short_can, kitchen_drawer]",
            "[End]"
        ],
        [
            "[Go to, drawer]",
            "[Open, kitchen_drawer]",
            "[Put, short_can, kitchen_drawer]",
            "[Close, drawer]",
            "[End]"
        ]
]
```

Here's an example to show how to calculate TP,

```
(1)  [Go to, kitchen](success)
(2)  [Open, drawer](success)
(3)  [Put, short_can, drawer](fail)
(4)  [Go to, kitchen_counter](success)
(5)  [Put, short_can, kitchen_counter](fail)
(6)  [Go to, drawer](success)
(7)  [Put, short_can, drawer](fail)
(8)  [Go to, kitchen_counter](success)
(9)  [Put, short_can, kitchen_counter](fail)
(10) [Go to, drawer](success)
(11) [Put, short_can, drawer](fail)
(12) [Go to, kitchen_counter](success)
(13) [Put, short_can, kitchen_counter](fail)
(14) [Go to, drawer](success)
(15) [Put, short_can, drawer](fail)
(16) [Go to, kitchen_counter](success)
(17) [Put, short_can, kitchen_counter](fail)
(18) [Go to, drawer](success)
(19) [Put, short_can, drawer](fail)
(20)  [Go to, kitchen_counter](success)
```

This is the result of one run, and we can see that the TP of this run is as calculated in section 2.2, $max \left( \frac{1}{3}, \frac{1}{4}, \frac{1}{2}, \frac{2}{5} \right) = 0.5$.

## C.2   SUCCESS END RATE

In the above result, the number of steps reach 20, and there is no $End$ action to terminate the task. Here's a example to show the success end.

```
(1)  [Go to, kitchen_counter](success)
(2)  [Go to, yellow_box](success)
(3)  [Pick, yellow_box](success)
(4)  [Go to, fridge](success)
(5)  [Put, yellow_box, fridge](fail)
(6)  [Open, fridge](fail)
(7)  [Go to, kitchen_counter](success)
(8)  [Put, yellow_box, kitchen_counter](success)
(9)  [Go to, fridge](success)
(10) [Open, fridge](success)
(11) [Go to, kitchen_counter](success)
(12) [Pick, yellow_box](success)
(13) [Go to, fridge](success)
(14) [Put, yellow_box, fridge](success)
(15) [Close, fridge](success)
(16) [End]
```

This is the result of one run for the task *take a yellow box and put it into the fridge*, and we can judge by its keypath that it complete the task successfully. It has $End$ action, so the $End$ is a success end which can be treated as one of the numerators when calculating SER in Section 2.2. In fact, as said in Section 2.2, successful task trajectory must have one end, but there maybe other unsuccessful task trajectories have ends, that's why we calculating SER.

## C.3   SUCCESS RE-PLAN RATE

First of all, the next action our agent takes after the previous action failed is called replan. Use the above subsection result as an example, and it's a successful task trajectory. In the step 5, the agent try

to put the yellow box in the fridge but failed, and then, it try to open the fridge which can be treated as a success replan even though it failed again. Since the action "open fridge" is a meaningful action which can lead to the final success. It's one of the numerators when calculating SRR in Section 2.2. Also, in the first subsection for TP, the example is an unseccessful task trajectory, so the actions like "put short can drawer" are not success replan.

# D    HIGH LEVEL PLANNING

In this section, we will should how the high-level planner described in Section 3.2 works step by step. To provide more intuitive understanding, we extract core sections from the original code and adapt them into a more general and easy-to-understand format to illustrate the process flow, this processing method is also applied to all subsequent code demonstrations. First, we provide the system information used in HomieBot, and all subsequent references to system information are consistent with what is provided here.

```
You are a powerful housework assistant, I will give you following information for you to make a
decision toward the final task.
(1) Observation images: Four first-person perspective images of the current environment, in the order
of front, left, back, and right.
(2) Task: Your final goal.
(3) Inventory: Your current assets, remember that you are a one-hand agent, which means you can't open
or pick when your Inventory is not None, and you can't put if your Inventory is None, this is very
important.
(4) Historical Execution: Subtasks that were already fulfilled in the history, and the execution status
 of each subtask(success or fail). You need to make decisions based on historical actions, current
circumstances and your final task.
(5) Feedback: Feedback will provide error information of the last execution, it will be None if the
last execution ends successfully.

You should output with following formats:
Analysis: Make a detailed summary of your current situation based on given information, analyse and
decide what to do next and output the reason of your decision.
Subtask: [action, target], choose your action from the action list [Go to, Pick, Put, Open, Close, End
], and the target can be a place or a object from your observation. If you choose Put as your action,
output in format [Put, object, place] which means put the object to the place. If the final task is
done and no more action is needed, just output [End].
Model: Choose one most suitable model in the model list [NoMaD, PixNav, octo, RT-1-X]. NoMaD can go to
a spot like living room, PixNav focuses on object navigation and can go to a object, octo can handle
with open and close, RT-1-X is good at picking and putting.

You need to focus on the consistency with previous subtasks. You should pay attention to current
Inventory and avoid conflicts.
Remember you can only go to the place and interact with the objects you observe in your sight.
Remember the logic between outputs, it is recommended to open the receptacle before you pick something
because you can't open while holding, and it's recommended to arrive the object place before you
interact with it.
Remember you just need to output the next subtask to be fulfilled and don't output a whole plan, this
is very important.
Remember you should output strictly with the response template.
Now, I will send the message so that you can make planning accordingly.
```

Next, we define some classes to make the overall process more readable and smooth. Here we only list most relevant and important parts in the process.

```python
import os
import json
import re

class Conversations:
    def __init__(self, max_round=20):
        self.system = SYSTEM_INFO
        self.history = []
        self.round = 0
        self.window = 3
        self.max_round = max_round

    def get_history_prompt(self):
        history_prompt = ""
        if self.round < self.window:
            history_prompt = "".join(self.history)
        else:
            history_prompt = "".join(self.history[-3:])
        return history_prompt

    def reset(self):
        self.history = []
        self.round = 0

    def save(self, save_path):
        with open(os.path.join(save_path, "conversation.json"), "w") as file:
            json.dump(self.history, file, indent=4)

class HomieBot:
```

```python
    def __init__(self):
        self.conv = Conversations()
        self.inventory = []
        self.comm = Communicator()

    def get_inventory(self):
        if len(self.inventory) == 0:
            return "None"
        else:
            return " ".join(self.inventory)

    def generate_instruction(self, task, feedback, historical_execution):
        if historical_execution == "":
            instruction = f"Task: {task}\nInventory: {self.get_inventory()}\nHistorical Execution: None
            \nFeedback: None\nNow based on the instruction above, please output Analysis, Subtask and
            Model in mentioned format.\n"
        else:
            instruction = f"Task: {task}\nInventory: {self.get_inventory()}\nHistorical Execution: {
            historical_execution}\nFeedback: {feedback}\nNow based on the instruction above, please
            output Analysis, Subtask and Model in mentioned format.\n"
        return instruction

    def update_inventory(self, subtask, feedback):
        subtask = subtask.lower()
        if "None" in feedback:
            if "pick" in subtask:
                obj = subtask.split.split(',')[1].strip()
                self.inventory.append(obj)
            if "put" in subtask:
                self.inventory.pop()
        else:
            if "put" in subtask and "the object is missing" in feedback:
                self.inventory.pop()

    def end(self):
        self.comm.close_connection()
```

the most important function *generate_instruction* works as described in Section 3.2, which contains *task*, *inventory*, *history* and *feedback*.

Afterward, we provide the process for HomieBot to execute the task in a single trajectory.

```python
homie = HomieBot()
task = "input your task"
save_path = "save_path"
feedback = ""
historical_execution = ""

while homie.conv.round < homie.conv.max_round:
    homie.conv.round += 1
    instruction = homie.generate_instruction(task, feedback, historical_execution)
    images = homie.comm.receive_env_images()

    output = model_inference(instruction, images)
    homie.conv.history.append(f"USER:\n{instruction}ASSISTANT:\n{output}\n")

    pattern = r'.*Analysis: *(.+?) *Subtask: *\[(.*?)\].*Model: *(.*?)$'
    match = re.search(pattern, output, re.DOTALL)
    analysis = match.group(1).strip()
    subtask = match.group(2).strip()
    model_choice = match.group(3).strip()

    homie.comm.send_subtask(subtask, model_choice, homie.get_inventory())
    feedback, signal = homie.comm.receive_feedback()

    homie.update_inventory(subtask, feedback)
    historical_execution += f"({homie.conv.round}) {subtask}({signal}) "

    if "end" in subtask.lower():
        break

homie.conv.save(save_path)
homie.end()
```

the realization of function *model_inference* varies from different models, but it's quite easy to deploy different models into HomieBot as we can see in the code.

# E  LOW LEVEL EXECUTION

## E.1  PIPELINE

```python
def error_detection(action, target, inventory, env):
    # Format Error Detection
    if action not in action_list:
        return 'fail', f'{action} is not in the action list! You should only choose actions in the list
.'

    mapping_dict = load_name_mapping()
    if target in mapping_dict:
        target = mapping_dict[target]
    else:
        return 'fail', f'{target} does not exist! Please choose another object'

    # Logical Error Detection
    if inventory != 'None' and action in ['pick', 'open', 'close']:
        return 'fail', f'Unable to {action}, the hand is full'
    if inventory == 'None' and action == 'put':
        return 'fail', f'Unable to {action}, the hand is empty'

    if action == 'put' and "closed" in check_status(target):
        return 'fail', f'Unable to put, the {target} is closed, you should open it first'

    if action in ['open','close'] and "non-interactive" in check_status(target):
        return 'fail', f'Can not {action} {target}! Please choose another object'

    # Distance Error Detection
    if action != "go to":
        distance = calculate_distance(env, target)
        if distance > 2:
            return 'fail', f'Unable to {action}, the target is far away'
        if distance < 0.1:
            return 'fail', f'Unable to {action}, the target is too close'

    return 'success', 'None'

max_count = 20
comm = Communicator()
save_path = "save_path"
count_steps = 1
env = init_env()

while count_steps <= max_count:
    images = get_env_images(save_path, env, count_steps)
    comm.send_env_images(images)

    action, target, inventory = comm.receive_subtask()
    if "end" in action.lower():
        comm.send_feedback("None", "success")
        break

    # Error Detection Before Execution
    signal, feedback = error_detection(action, target, inventory, env)
    if signal == "fail":
        comm.send_feedback(feedback, signal)
        break

    for retry in range(3):
        reset_arm(env)
        # Error Detection During and After Execution
        signal, feedback, env = execution(action, target, inventory, env)
        if signal == 'success':
            break
        elif action == 'put' and env['grasped_obj'] is None:
            feedback = f'Unable to {action}, and the object is missing'
            break
        elif retry == 2:
            feedback = f'Unable to {action}, the subtask is too difficult to perform'
    if signal == 'success':
        feedback = "None"

    count_steps += 1
    comm.send_feedback(feedback, signal)
```

## E.2  SKILLS

The skill we choose and their functions are shown in Table E1.

## E.3  MODELS

**M3**    (Gu et al., 2022) can flexible interact with target objects from various locations based on the integration of manipulative skills and mobility, while navigational skills are designed to accommodate multiple endpoints, ultimately leading to successful operations. Specifically, M3 implements these concepts by emphasizing mobile manipulation skills over fixed skills and training navigational skills using area targets rather than point targets.

Table E1: The list of skills we used with descriptions and examples

| Skill | Description | Example |
|---|---|---|
| Pick object | Pick an object up | pick sugar box |
| Put object to place | Put an object into a place | put lemon on brown table |
| Open container | Open the container | open the fridge |
| Close container | Close the container | close the kitchen drawer |
| Go to place | navigate to a place | navigate TV stand |
| Go to object | navigate to where an object is | navigate bowl |
| End | End the execution | End |

Table E2: Descriptions of Low Level Models used in HOMIEBOT.

| Model | Input | Capability | Task |
|---|---|---|---|
| RT-1-X(Brohan et al., 2022) | RGB & Instructions | Manipulation | Picking & Placing |
| Octo(Team et al., 2024b) | RGB & Instructions | Manipulation | Opening & Closing |
| NoMaD(Sridhar et al., 2024) | RGB & Goal-Image | Image-Navigation | Navigate to Spot & Large Object |
| PixNav(Cai et al., 2024) | RGB & Goal-Name | Pixel-Navigation | Navigate to Object |

**RT-1-X** ( (Padalkar et al., 2023)) architecture utilizes image and text instructions as inputs, and generates discrete end-effector actions as outputs. Specifically, RT-1-X is a transformer-based model that guides robotic arms to complete various manipulation tasks. RT-1-X is an extension of the RT-1 ( (Brohan et al., 2022)) model, which is designed for robot control and trained on a large-scale robot dataset.

**Octo** ( (Team et al., 2024b)) is an open-source, general-purpose policy for robotic manipulation based on transformers. It supports flexible task and observation definition and can be quickly integrated into new observation and action spaces.

**NoMaD** ( (Sridhar et al., 2024)) trains a single diffusion strategy for goal-oriented navigation and goal-independent exploration, the first one is to reach user-specified goals after localization and the second one is to search new environments. The method is instantiated using a transformer-based large-scale policy trained on data from various ground robots.

**PixNav** ( (Cai et al., 2024)) is a pixel-guided navigational skill. It designs an LLM-based planner that utilizes common sense between objects and rooms to select the optimal waypoints, which are then executed by a pixel navigation strategy to achieve long-line-of-sight navigation. In this pipeline, we use its ability of finding the optimal waypoint and pixel navigation to navigate to some specific small object such as lemon and sugar box.

### E.4 ERROR CLASSIFICATION

**Logical error.** If the hand already has an object (inventory is not empty) but still attempts to perform a pick/open/close operation, the execution will fail, and the message "the hand is full" will be returned; if the hand has no object (inventory is empty) but still attempts to perform a place operation, the execution will fail, and the message "the hand is empty" will be returned; if the item is not a container but still attempts to perform a open/close operation, the execution will fail, and the message "please choose another object" will be returned. In the execution with environment state information, if the container is closed and a place operation is still attempted, the execution will fail, and the message "the container is closed, you should open it first" will be returned.

**Distance error.** In the execution with environment state information, if the agent is too close to the target, causing the arm to be unable to extend properly but still attempts to perform a pick/place/open/-close operation, the execution will fail, and the message "the target is too close" will be returned; if the agent is too far from the target, causing it to be unable to reach the target object but still attempts to perform a pick/place/open/close operation, the execution will fail, and the message "the target is far away" will be returned.

**Format Error.** For high level planning, it may output an object which is not in the scene, that is, in low level execution, we can't find an object with a name matching the input in the scene, the message "please choose another object" will be returned; also, high level planning may output in a wrong operation which can not be performed, the message "You should only choose actions in the list" will be returned.

**Execution Error** Due to the limited capabilities of low-level models, sometimes the failure is not caused by HLP. Therefore, each action can be executed up to three times. If it fails after three times, it will return a message "the subtask is too difficult to perform"; also, when performing a put operation, if the agent put the wrong place, it will return a message "the object is missing" to remind the agent to re-plan and re-pick.

# F  DATA AUGMENTATION

## F.1  SFT AUGMENTATION

To expand the original dataset size, we first use GPT-4o (Achiam et al., 2023) to regenerate text descriptions. Here is the regeneration code clip, we just show how to regenerate task descriptions, but the regeneration of subtask analysis uses the same template.

```python
client = OpenAI(api_key='')
completion = client.chat.completions.create(
    model="gpt-4o",
    messages=[
        {"role": "system", "content": "Rewrite the following text with the same meaning but in a
        different description while do not change object's name: "},
        {"role": "user", "content": task}
    ]
)
```

Next we show how to convert a single EMMOE data into fix-format conversation data. After processing, each individual subtask will be combined with all previously subtasks to form a SFT data.

```python
import os
with open(task_path) as file:
    content = file.read()

content = content.split("\n\n")
task = content[0]
historical = ""
sft_data = []

for i, subtask_info in enumerate(content[1:]):
    subtask_data = {}
    subtask_info = subtask_info.strip().split("\n")
    if subtask_info[0] == '':
        continue
    subtask_id, decision = subtask_info[0].split(': ')
    subtask_id = subtask_id.lower()
    analysis = subtask_info[1]

    if "End" not in decision:
        action, model_choice = decision.strip(')').split(' (')
    else:
        action = "[End]"
        model_choice = "None"

    image_paths = [
        os.path.join(save_dir, f"{subtask_id}_front.png"),
        os.path.join(save_dir, f"{subtask_id}_left.png"),
        os.path.join(save_dir, f"{subtask_id}_back.png"),
        os.path.join(save_dir, f"{subtask_id}_right.png")
    ]
    for path in image_paths:
        if not os.path.exists(path):
            raise FileNotFoundError(f"File does NOT exist: {path}")
    if i == 0:
        instruction = f"{task}\nInventory: None\nHistorical Execution: None\nFeedback: None\nNow,
        please output Analysis, Subtask and Model, according to the instruction above."
    else:
        instruction = f"{task}\n{inventory}\nHistorical Execution:{historical}\n{feedback}\nNow, please
         output Analysis, Subtask and Model, according to the instruction above."
    answer = f"{analysis}\nSubtask: {action}\nModel: {model_choice}"

    feedback = subtask_info[2]
    inventory = subtask_info[3]
    if "None" in feedback:
        historical += f"({i+1}){decision} (success)\n"
```

```
        else:
            historical += f"({i+1}){decision} (fail)\n"

        conv = []
        conv.append({"from": "human", "value": instruction})
        conv.append({"from": "gpt", "value": answer})

        task_id = '_'.join(task_path.split('.')[0].split('/')[-2:])

        subtask_data["id"] = task_id + '_' + subtask_id
        subtask_data["image"] = image_paths
        subtask_data["conversations"] = conv

        sft_data.append(subtask_data)
```

We also provide some data samples for more intuitive understanding.

```
[
    {
        "id": "21_info_subtask13",
        "image": [
            "./HomieBot/data/train/21/subtask13_front.png",
            "./HomieBot/data/train/21/subtask13_left.png",
            "./HomieBot/data/train/21/subtask13_back.png",
            "./HomieBot/data/train/21/subtask13_right.png"
        ],
        "conversations": [
            {
                "from": "human",
                "value": "Task: clean up the blue sofa\nInventory: None\nHistorical Execution:(1)[Go to
                , blue_sofa] (PixNav) (success)\n(2)[Pick, cracker_box] (RT-1-X) (success)\n(3)[Go to,
                kitchen_counter] (NoMaD) (success)\n(4)[Put, cracker_box, kitchen_counter] (RT-1-X) (
                success)\n(5)[Go to, blue_sofa] (PixNav) (success)\n(6)[Pick, sugar_box] (RT-1-X) (
                success)\n(7)[Go to, kitchen_counter] (NoMaD) (success)\n(8)[Put, sugar_box,
                kitchen_counter] (RT-1-X) (success)\n(9)[Go to, blue_sofa] (PixNav) (success)\n(10)[
                Pick, soup_can] (RT-1-X) (success)\n(11)[Go to, kitchen_counter] (NoMaD) (success)\n
                (12)[Put, soup_can, kitchen_counter] (RT-1-X) (success)\n\nFeedback: None\nNow, please
                output Analysis, Subtask and Model, according to the instruction above."
            },
            {
                "from": "gpt",
                "value": "Analysis: I successfully put the soup_can on the kitchen_counter, according
                to my task, I should go back to the sofa to check whether it's clean or not\nSubtask: [
                Go to, blue_sofa]\nModel: PixNav"
            }
        ]
    },
    {
        "id": "21_info_subtask14",
        "image": [
            "./HomieBot/data/train/21/subtask14_front.png",
            "./HomieBot/data/train/21/subtask14_left.png",
            "./HomieBot/data/train/21/subtask14_back.png",
            "./HomieBot/data/train/21/subtask14_right.png"
        ],
        "conversations": [
            {
                "from": "human",
                "value": "Task: clean up the blue sofa\nInventory: None\nHistorical Execution:(1)[Go to
                , blue_sofa] (PixNav) (success)\n(2)[Pick, cracker_box] (RT-1-X) (success)\n(3)[Go to,
                kitchen_counter] (NoMaD) (success)\n(4)[Put, cracker_box, kitchen_counter] (RT-1-X) (
                success)\n(5)[Go to, blue_sofa] (PixNav) (success)\n(6)[Pick, sugar_box] (RT-1-X) (
                success)\n(7)[Go to, kitchen_counter] (NoMaD) (success)\n(8)[Put, sugar_box,
                kitchen_counter] (RT-1-X) (success)\n(9)[Go to, blue_sofa] (PixNav) (success)\n(10)[
                Pick, soup_can] (RT-1-X) (success)\n(11)[Go to, kitchen_counter] (NoMaD) (success)\n
                (12)[Put, soup_can, kitchen_counter] (RT-1-X) (success)\n(13)[Go to, blue_sofa] (PixNav
                ) (success)\n\nFeedback: None\nNow, please output Analysis, Subtask and Model,
                according to the instruction above."
            },
            {
                "from": "gpt",
                "value": "Analysis: According to the historical execution and final task, I have
                finally finished the task and it's time to end\nSubtask: [End]\nModel: None"
            }
        ]
    }
]
```

## F.2 DPO AUGMENTATION

The last step for SFT Augmentation and the first step of DPO Augmentation occur simultaneously, failed subtasks are considered junk data for SFT but can be used to construct DPO data, so we run $data\_sift$ function first. After getting initial DPO data, we will use SFT data, $model\_change$, $order\_change$ and $action\_change$ to construct final DPO dataset.

```python
def data_sift(subtask_list):
    sft_data = []
    dpo_data = []
    flag = 1
    for i in range(1, len(subtask_list)):
        if "Feedback: None" in subtask_list[i]["conversations"][0]["value"]:
            sft_data.append(subtask_list[i-1])
            if flag == 0:
                dpo_data.append({
                    "prompt": subtask_list[i-2]["conversations"][0]["value"],
                    "chosen": '\n'.join(subtask_list[i-1]["conversations"][1]["value"].split('\n')[1:])
                    ,
                    "rejected": '\n'.join(subtask_list[i-2]["conversations"][1]["value"].split('\n')
                    [1:])
                })
            flag = 1
        else:
            flag = 0
    sft_data.append(subtask_list[-1])

    return sft_data, dpo_data

def dpo_augment(sft_data, dpo_data):
    for i in range(len(sft_data)):
        prompt = sft_data[i]["conversations"][0]["value"]
        chosen = '\n'.join(sft_data[i]["conversations"][1]["value"].split('\n')[1:])
        if "End" in sft_data[i]["conversations"][1]["value"]:
            continue

        def model_change(chosen):
            if "NoMaD" in chosen:
                return chosen.replace("NoMaD", "PixNav")
            elif "PixNav" in chosen:
                return chosen.replace("PixNav", "NoMaD")
            elif "octo" in chosen:
                return chosen.replace("octo", "RT-1-X")
            else:
                return chosen.replace("RT-1-X", "octo")

        def order_change(i, sft_data):
            return '\n'.join(sft_data[i+1]["conversations"][1]["value"].split('\n')[1:])

        def action_change(chosen):
            if "Pick" in chosen:
                return chosen.replace("Pick", "Fetch")
            elif "Put" in chosen:
                return chosen.replace("Put", "Place")
            elif "Go to" in chosen:
                return chosen.replace("Go to", "Move")
            elif "Open" in chosen:
                return chosen.replace("Open", "Pull")
            elif "Close" in chosen:
                return chosen.replace("Close", "Push")

        reject1 = model_change(chosen)
        reject2 = order_change(i, sft_data)
        reject3 = action_change(chosen)
        dpo_data.append({"prompt": prompt, "chosen": chosen, "rejected": reject1})
        dpo_data.append({"prompt": prompt, "chosen": chosen, "rejected": reject2})
        dpo_data.append({"prompt": prompt, "chosen": chosen, "rejected": reject3})

    return dpo_data
```

Notably, action $End$ is special among all available actions and it will only appear as $rejected$ in DPO data. In the first augmentation stage and $order\_change$, since the relationship between $chosen$ and $rejected$ is $O_i$ and $O_{i+1}$ (see definitions in Section 4.1) and there are no other subtasks after $End$, which means other actions might appear in either $chosen$ or $rejected$ while $End$ can only be the $rejected$. But this effect of suppressing the $End$ output is exactly what we want. Even executing a few extra steps after completing the task is better than terminating early without finishing the task. That is to say, We hope the model could consider more and do not output $End$ so easily. Experimental results in Table 2 and Table 3 confirm the effectiveness of this method as we can see an improvement in $SER$ metric, another positive phenomenon in results is that the length of the successful paths hasn't increased significantly as we observe in $PLWSR$ and $TP$.

Finally, we provide some DPO data examples.

```json
[
    {
        "prompt": "Task: Clear everything off the table in front of you and place all the items in the
        sink.\nInventory: None\nHistorical Execution:(1)[Pick, yellow_box] (RT-1-X) (success)\n(2)[Put,
        yellow_box, sink] (RT-1-X) (success)\n\nFeedback: None\nNow, please output Analysis, Subtask
        and Model, according to the instruction above.",
        "chosen": "Subtask: [Go to, red_can]\nModel: PixNav",
        "rejected": "Subtask: [Pick, red_can]\nModel: RT-1-X"
    },
```

```
    {
        "prompt": "Task: Collect all the fruit located on the brown table and place them on the sofa.\
        nInventory: None\nHistorical Execution:(1)[Go to, brown_table] (NoMaD) (success)\n(2)[Pick,
        orange] (RT-1-X) (success)\n(3)[Go to, sofa] (PixNav) (success)\n(4)[Put, orange, sofa] (RT-1-X
        ) (success)\n(5)[Go to, brown_table] (NoMaD) (success)\nFeedback: None\nNow, please output
        Analysis, Subtask and Model, according to the instruction above.",
        "chosen": "Subtask: [Pick, pear]\nModel: RT-1-X",
        "rejected": "Subtask: [Fetch, pear]\nModel: RT-1-X"
    },
    {
        "prompt": "Task: find a blue can for me\nInventory: None\nHistorical Execution: None\nFeedback:
         None\nNow, please output Analysis, Subtask and Model, according to the instruction above.",
        "chosen": "Subtask: [Go to, fridge]\nModel: PixNav",
        "rejected": "Subtask: [Go to, fridge]\nModel: NoMaD"
    }
]
```

# G  TRAINING DETAILS

## G.1  TRAINING PARAMETERS

We use Video-LLaVA-7B (Zhang et al., 2023) as our base model, we also use the training scripts they provide and partial parameters for $sft$ are as follows.

```
--lora_enable True
--lora_r 128
--lora_alpha 256
--mm_projector_lr 2e-5
--bits 4
--mm_projector_type mlp2x_gelu
--mm_vision_select_layer -2
--mm_use_im_start_end False
--mm_use_im_patch_token False
--image_aspect_ratio pad
--group_by_modality_length True
--bf16 True
--num_train_epochs 1
--per_device_train_batch_size 16
--per_device_eval_batch_size 4
--gradient_accumulation_steps 1
--evaluation_strategy "no"
--save_strategy "steps"
--save_steps 50000
--save_total_limit 1
--learning_rate 5e-4
--weight_decay 0.
--warmup_ratio 0.03
--lr_scheduler_type "cosine"
--logging_steps 1
--tf32 True
--model_max_length 2048
--tokenizer_model_max_length 3072
--gradient_checkpointing True
--dataloader_num_workers 4
--lazy_preprocess True
--report_to tensorboard
```

We use finetuned model as our base and reference model, and use open-source $trl$ package and parameters for $dpo$ are as follows.

```
bnb_config = BitsAndBytesConfig(
    load_in_4bit=True,
    bnb_4bit_compute_dtype=torch.float16,
    bnb_4bit_use_double_quant=True,
    bnb_4bit_quant_type='nf4'
)
training_args = DPOConfig(
    per_device_train_batch_size=16,
    per_device_eval_batch_size=4,
    gradient_accumulation_steps=1,
    gradient_checkpointing=True,
    max_grad_norm=0.3,
    num_train_epochs=1,
    save_steps=1000,
    learning_rate=5e-6,
    bf16=True,
    save_total_limit=1,
    logging_steps=10,
    output_dir=output_dir,
    optim="paged_adamw_32bit",
    lr_scheduler_type="cosine",
    warmup_ratio=0.03,
    remove_unused_columns=False
)
```

```
peft_config = LoraConfig(
    r=8,
    lora_alpha=8,
    target_modules=find_all_linear_names(model),
    lora_dropout=0.05,
    bias="none",
    task_type="CAUSAL_LM",
)
dpo_trainer = DPOTrainer(
    model,
    model_ref,
    args=training_args,
    beta=0.1,
    train_dataset=train_dataset,
    eval_dataset=eval_dataset,
    tokenizer=tokenizer,
    max_prompt_length=2048,
    max_length=2048,
)
```

## H EXPERIMENTAL DETAILS

### H.1 BASELINE SETUP

To make it more convenient for different models to deploy into our system without training, we slightly lower output format requirements, here shows the adapatations.

```
import re

pattern = r'.*Analysis: *(.+?) *Subtask: *\[(.*?)\].*Model: *(.*?)$'
match = re.search(pattern, output, re.DOTALL)
if match == None:
    pattern = r'.*Analysis: *(.+?) *Subtask: *(.*?) *Model: *(.*?)$'
    match = re.search(pattern, output, re.DOTALL)
```

Despite lowering the output format standards, the output from 7B-sized models still fails to meet our least requirements. They either do not output single-step subtasks or the subtask format is far from requirements. This issue is difficult to resolve by merely adjusting prompts. Therefore, we leverage the in-context learning abilities of these models by providing an output template example before each inference. Here, we provide the inference template for Qwen2-VL (Wang et al., 2024) MiniCPM-V 2.6 (Yao et al., 2024) respectively.

Qwen2VL

```
messages = [
        {"role": "system", "content": homie.conv.system},
        {"role": "user",
         "content": "here is an example output, please strictly follow its format and system reminders
         in your output:\nAnalysis: According to my final task, I need to fetch apples first, but it's
         a better choice to go the fridge and open it first, which will avoid potential conflicts, so I
          should go to the fridge next\nSubtask: [Go to, fridge]\nModel: NoMaD\n",
        },
        {"role": "assistant",
         "content": "I will surely follow the given format, now you can send prompt to me."
        },
        {"role": "user",
         "content": [
                {"type": "image", "image": images[0]},
                {"type": "image", "image": images[1]},
                {"type": "image", "image": images[2]},
                {"type": "image", "image": images[3]},
                {"type": "text", "text": instruction}]
        }
]
prompt = processor.apply_chat_template(
        messages, tokenize=False, add_generation_prompt=True
)
image_inputs, video_inputs = process_vision_info(messages)
inputs = processor(
        text=[prompt],
        images=image_inputs,
        videos=video_inputs,
        padding=True,
        return_tensors="pt"
).to("cuda")
generated_ids = model.generate(**inputs, max_new_tokens=512)
enerated_ids_trimmed = [
        out_ids[len(in_ids) :] for in_ids, out_ids in zip(inputs.input_ids, generated_ids)
]
outputs = processor.batch_decode(
        generated_ids_trimmed, skip_special_tokens=True, clean_up_tokenization_spaces=False
```

```
        )
```

MiniCPM-V 2.6

```
image_loads = [Image.open(image).convert('RGB') for image in images]
messages = [
        {"role": "user",
         "content": "here is an example output, please strictly follow its format and system reminders
         in your output:\nAnalysis: According to my final task, I need to fetch apples first, but it's
         a better choice to go the fridge and open it first, which will avoid potential conflicts, so I
          should go to the fridge next\nSubtask: [Go to, fridge]\nModel: NoMaD\n",
        },
        {"role": "assistant",
         "content": "I will surely follow the given format, now you can send prompt to me.",
        },
        {"role": "user",
         "content": [image_loads[0], image_loads[1], image_loads[2], image_loads[3], instruction]
        }
]

output = model.chat(
        image=None,
        system_prompt=homie.conv.system,
        tokenizer=tokenizer
)
```

## H.2 RESULTS

Here we provide more detailed results of experiments in Section 4.4. Table H3 and Table H4 show the statistics results in percentages while Table H5 and Table H6 show original counts. Table H7 show the original counts and success rate range of each action.

Table H3: **Successful Trajectories Error Statistics** All definitions are same as in Section 4.4. Additionally, we add statistics of four primary types.

| Models | L1 | L2 | L3 | L4 | L | D1 | D2 | D | F1 | F2 | F | E1 | E2 | E | All |
|---|---|---|---|---|---|---|---|---|---|---|---|---|---|---|---|
| GPT-4o(Achiam et al., 2023) | 3.97 | 0.79 | 0.79 | 0 | 5.56 | 44.44 | 0 | 44.44 | 1.59 | 17.46 | 19.05 | 15.87 | 15.08 | 30.95 | 30.29 |
| Gemini-1.5-Pro(Team et al., 2024a) | 3.85 | 3.85 | 0 | 7.69 | 15.38 | 48.08 | 0 | 48.08 | 0 | 17.31 | 17.31 | 15.38 | 3.85 | 19.23 | 21.80 |
| Qwen2-VL-7B(Wang et al., 2024) | 0 | 0 | 0 | 0 | 0 | 100 | 0 | 0 | 0 | 0 | 0 | 0 | 0 | 0 | 20 |
| MiniCPM-V 2.6(Yao et al., 2024) | 0 | 0 | 0 | 0 | 0 | 100 | 0 | 0 | 0 | 0 | 0 | 0 | 0 | 0 | 6.67 |
| HomieBot-7B (SFT) | 10.53 | 9.77 | 12.78 | 1.50 | 34.59 | 36.09 | 0 | 36.09 | 0 | 3.01 | 3.00 | 24.06 | 2.26 | 26.32 | 14.41 |
| HomieBot-7B (SFT+DPO) | 10.17 | 15.25 | 9.32 | 3.39 | 38.14 | 33.05 | 0 | 33.05 | 0 | 3.39 | 3.39 | 25.42 | 0 | 25.42 | 12.87 |

Table H4: **Failed Trajectories Error Statistics**

| Models | L1 | L2 | L3 | L4 | L | D1 | D2 | D | F1 | F2 | F | E1 | E2 | E | All |
|---|---|---|---|---|---|---|---|---|---|---|---|---|---|---|---|
| GPT-4o(Achiam et al., 2023) | 6.87 | 0.12 | 0.69 | 3.65 | 11.34 | 8.41 | 0.06 | 8.47 | 0.57 | 64.88 | 65.45 | 13.99 | 0.75 | 14.74 | 73.61 |
| Gemini-1.5-Pro(Team et al., 2024a) | 7.48 | 1.52 | 2.41 | 6.45 | 17.86 | 9.41 | 0 | 9.41 | 0 | 47.86 | 47.86 | 22.76 | 2.10 | 24.86 | 68.38 |
| Qwen2-VL-7B(Wang et al., 2024) | 2.17 | 9.49 | 0.99 | 3.56 | 16.21 | 7.71 | 0 | 7.71 | 4.74 | 54.35 | 59.09 | 16.40 | 0.59 | 17.00 | 27.74 |
| MiniCPM-V 2.6(Yao et al., 2024) | 8.58 | 0.80 | 0.92 | 1.72 | 12.01 | 7.78 | 0 | 7.78 | 3.49 | 65.39 | 68.88 | 10.87 | 0.46 | 11.33 | 31.08 |
| HomieBot-7B (SFT) | 11.31 | 23.85 | 9.86 | 4.20 | 49.24 | 11.77 | 0 | 11.77 | 0.61 | 11.47 | 12.08 | 24.54 | 2.37 | 26.91 | 35.70 |
| HomieBot-7B (SFT+DPO) | 11.46 | 23.90 | 11.13 | 2.62 | 49.10 | 9.25 | 0 | 9.25 | 0.25 | 17.27 | 17.51 | 22.67 | 1.47 | 24.14 | 35.88 |

Table H5: **Original Successful Trajectories Statistics** All data are integers.

| Models | L1 | L2 | L3 | L4 | L | D1 | D2 | D | F1 | F2 | F | E1 | E2 | E | All |
|---|---|---|---|---|---|---|---|---|---|---|---|---|---|---|---|
| GPT-4o(Achiam et al., 2023) | 5 | 1 | 1 | 0 | 7/126 | 56 | 0 | 56/126 | 2 | 22 | 24/126 | 20 | 19 | 39/126 | 126/416 |
| Gemini-1.5-Pro(Team et al., 2024a) | 4 | 4 | 0 | 8 | 16/104 | 50 | 0 | 50/104 | 0 | 18 | 18/104 | 16 | 4 | 20/104 | 104/477 |
| Qwen2-VL-7B(Wang et al., 2024) | 0 | 0 | 0 | 0 | 0/9 | 9 | 0 | 9/9 | 0 | 0 | 0/9 | 0 | 0 | 0/9 | 9/45 |
| MiniCPM-V 2.6(Yao et al., 2024) | 0 | 0 | 0 | 0 | 0/1 | 1 | 0 | 0/1 | 0 | 0 | 0/1 | 0 | 0 | 0/1 | 1/15 |
| HomieBot-7B (SFT) | 14 | 13 | 17 | 2 | 46/133 | 48 | 0 | 48/133 | 0 | 4 | 4/133 | 32 | 3 | 35/133 | 133/923 |
| HomieBot-7B (SFT+DPO) | 12 | 18 | 11 | 4 | 45/118 | 39 | 0 | 39/118 | 0 | 4 | 4/118 | 30 | 0 | 30/118 | 118/917 |

# I CASE STUDY

We show case studies of the inference from HomieBot in various situations as follows.

**Case 1: Successful trajectory**

Here we show a successful trajectory of our HomieBot (DPO version). To facilitate understanding, we convert the dialogue data into the original EMMOE data format. As shown, even if errors occur, timely adjustments can be made through feedback, ensuring the correctness of the execution process.

Table H6: **Original Failed Trajectories Statistics**

| Models | L1 | L2 | L3 | L4 | L | D1 | D2 | D | F1 | F2 | F | E1 | E2 | E | All |
|---|---|---|---|---|---|---|---|---|---|---|---|---|---|---|---|
| GPT-4o(Achiam et al., 2023) | 228 | 4 | 23 | 121 | 376/3317 | 279 | 2 | 281/3317 | 19 | 2152 | 2171/3317 | 464 | 25 | 489/3317 | 3317/4506 |
| Gemini-1.5-Pro(Team et al., 2024a) | 217 | 44 | 70 | 187 | 518/2900 | 273 | 0 | 273/2900 | 0 | 1388 | 1388/2900 | 660 | 61 | 721/2900 | 2900/4241 |
| Qwen2-VL-7B(Wang et al., 2024) | 33 | 144 | 15 | 54 | 246/1518 | 117 | 0 | 117/1518 | 72 | 825 | 897/1518 | 249 | 9 | 258/1518 | 1518/5472 |
| MiniCPM-V 2.6(Yao et al., 2024) | 150 | 14 | 16 | 30 | 210/1748 | 136 | 0 | 136/1748 | 61 | 1143 | 1204/1748 | 190 | 8 | 198/1748 | 1748/5624 |
| HomieBot-7B (SFT) | 148 | 312 | 129 | 55 | 644/1308 | 154 | 0 | 154/1308 | 8 | 150 | 158/1308 | 321 | 31 | 352/1308 | 1308/3664 |
| HomieBot-7B (SFT+DPO) | 140 | 292 | 136 | 32 | 600/1222 | 113 | 0 | 113/1222 | 3 | 211 | 214/1222 | 277 | 18 | 295/1222 | 1222/3406 |

Table H7: Count are the number of each action error with a total count behind. $SR_{range}$ are percentages to indicate the range of success rates of each action, with the average value shown in parentheses.

| Metrics | Go to | Pick | Place | Open | Close |
|---|---|---|---|---|---|
| Count | 938/2437 | 1213/2437 | 178/2437 | 81/2437 | 27/2437 |
| $SR_{range}$ | (45.32)31.19~82.89 | (22.45)18.34~28.95 | (40.97)29.06~69.40 | (43.13)30.13~75.84 | (36.45)26.72~57.37 |

```
Task: put the green_can on the brown_table to the fridge

#Round1
Analysis: According to my final task, I need to go to the green_can first.
Subtask: [Go to, green_can]
Inventory: None
Feedback: None

#Round2
Analysis: I successfully arrived the green_can and I should pick it next
Subtask: [Pick, green_can]
Inventory: green_can
Feedback: None

#Round3
Analysis: I successfully picked the green_can, so according to my final task, I should go to the fridge
 to put it down.
Subtask: [Go to, fridge]
Inventory: green_can
Feedback: None

#Round4
Analysis: I successfully arrived the fridge and I should put the green_can into it next
Subtask: [Put, green_can, fridge]
Inventory: green_can
Feedback: Unable to put, the fridge is closed, you should open it first

#Round5
Analysis: I failed to put the green_can into the fridge, since the fridge is closed, so I should open
the fridge first
Subtask: [Open, fridge]
Inventory: green_can
Feedback: Unable to open, the hand is full

#Round6
Analysis: I failed to open the fridge because my hand is full, so I should put the green_can on the
brown_table first
Subtask: [Go to, brown_table]
Inventory: green_can
Feedback: None

#Round7
Analysis: I successfully arrive the brown_table and I should put the green_can on it
Subtask: [Put, green_can, brown_table]
Inventory: None
Feedback: None

#Round8
Analysis: I successfully put the green_can on the brown_table, so I should go to the fridge next
Subtask: [Go to, fridge]
Inventory: None
Feedback: None

#Round9
Analysis: I successfully arrived the fridge and I should open it next
Subtask: [Open, fridge]
Inventory: None
Feedback: None

#Round10
Analysis: I successfully open the fridge and find the green_can, so I should go to the green_can.
Subtask: [Go to, green_can]
Inventory: None
Feedback: None

#Round11
```

```
Analysis: I successfully arrive the green_can and find it, so I should pick it.
Subtask: [Pick, green_can]
Inventory: green_can
Feedback: None

#Round12
Analysis: I successfully picked the green_can, so according to my final task, I should go to the fridge
 to put it down.
Subtask: [Go to, fridge]
Inventory: green_can
Feedback: None

#Round13
Analysis: I successfully arrived the fridge and I should put the green_can into it next
Subtask: [Put, green_can, fridge]
Inventory: None
Feedback: None

#Round14
Analysis: I successfully put the green_can into the fridge, so I should close the fridge next
Subtask: [Close, fridge]
Inventory: None
Feedback: None\n

#Round15
Analysis: According to the historical execution and final task, I have finally finished the task and it
's time to end
Subtask: [End]
Inventory: None
Feedback: None
```

## Case 2: Terrible Grounding

Here we show a terrible grounding problem issues during the inference of GPT4-o (Achiam et al., 2023). Due to the lengthy path, we only highlight the most critical subtask outputs and their execution results to emphasize errors (following cases are handled similarly). We can see that the ungrounded output directly prevents the process to be continued. Even after informing the model that the object doesn't exist, the issue remains unresolved. The model continues to output incorrect objects or makes mistakes again after a few steps.

```
Task: clean up the beige table and put all items to kitchen

(1) Go to, beige table(success)
(2) Pick, item(fail)
(3) Pick, item(fail)
(4) Pick, item(fail)
(5) Pick, item(fail)
(6) Pick, item(fail)
(7) Pick, item(fail)
(8) Pick, item on the table(fail)
(9) Pick, item on the table(fail)
(10) Pick, item on the table(fail)
(11) Pick, item on the table(fail)
(12) Pick, item(fail)
(13) Pick, item on the table(fail)
(14) Pick, item on the beige table(fail)
(15) Pick, stack of papers(fail)
(16) Pick, box(fail)
(17) Go to, closer to the table(fail)
(18) Pick, item on the table(fail)
(19) Pick, visible item on the beige table(fail)
```

## Case 3: Limited LLE

Here we show a problem issues from the limited ability of low-level models. As we can see in step7 and step8, though high-level planner makes correct plans, execution still fails due to the limited ability of low-level models, this problem occurs repeatedly during whole process and finally leads to the failue the trajectory.

```
Task: take a yellow box and put it into the fridge.

(1) Go to, living room(success)
(2) Pick, yellow box(success)
(3) Go to, kitchen(success)
(4) Open, fridge(fail)
(5) Put, yellow box, counter(fail)
(6) Put, yellow box, table(fail)
(7) Go to, fridge(success)
(8) Open, fridge(fail)
(9) Put, yellow box, counter(fail)
(10) Open, fridge(fail)
(11) Put, yellow box, counter(fail)
(12) Open, fridge(fail)
(13) Put, yellow box, counter(fail)
```

```
(14) Open, fridge(fail)
(15) Put, yellow box, counter(fail)
(16) Open, fridge(fail)
(17) Put, yellow box, counter(fail)
(18) Open, fridge(fail)
(19) Put, yellow box, counter(fail)
```

### Case 4: Meaningless Outputs

Here we show a case of Qwen2-VL (Wang et al., 2024) generating a large amount of meaningless outputs during the inference. As we can see, even though all subtasks are successful, the agent keep circling in place without making progress. These meaningless outputs quickly consume the remaining execution steps, ultimately causing the task to fail.

```
Task: find two bowls in the room and put them in the kitchen sink
(1) Go to, kitchen(success)
(2) Go to, kitchen(success)
(3) Go to, kitchen(success)
(4) Go to, kitchen(success)
(5) Go to, kitchen(success)
(6) Go to, kitchen(success)
(7) Go to, living room(success)
(8) Go to, living room(success)
(9) Go to, kitchen(success)
(10) Go to, kitchen(success)
(11) Go to, living room(success)
(12) Go to, living room(success)
(13) Go to, living room(success)
(14) Go to, living room(success)
(15) Go to, living room(success)
(16) Go to, living room(success)
(17) Go to, living room(success)
(18) Go to, living room(success)
(19) Go to, living room(success)
```

### Case 5: Solvable D1 Error

Here we show how $D1$ error is solvable during the inference of Gemini-1.5-Pro (Team et al., 2024a). As we can see in step3, after a $D1$ error happens, a $Go\ to$ action can effectively solve it and facilitate the success of the whole trajectory.

```
Task: fetch one crack box and one sugar box and put them on the beige table

(1) [Go to, beige_table](success)
(2) [Pick, cracker_box](fail)
(3) [Go to, cracker_box](success)
(4) [Pick, cracker_box](success)
(5) [Go to, beige_table](success)
(6) [Put, cracker_box, beige_table](success)
(7) [Go to, brown_table](success)
(8) [Put, sugar_box, brown_table](fail)
(9) [Go to, beige_table](success)
(10) [Put, sugar_box, beige_table](fail)
(11) [Go to, sugar_box](success)
(12) [Pick, sugar_box](success)
(13) [Go to, beige_table](success)
(14) [Put, sugar_box, beige_table](success)
(15) [End](success)
```

