# OpenReview forum: "HomieBot: an Adaptive System for Embodied Mobile Manipulation in Open Environments"
_ICLR.cc/2025/Conference — ICLR 2025 Conference Withdrawn Submission_

### Official Review · Reviewer_tYVZ · 2024-10-31

**Soundness:** 2
**Presentation:** 1
**Contribution:** 1
**Rating:** 3
**Confidence:** 4

**Summary:**

The paper introduces a unified benchmark named “EMMOE” designed to simultaneously evaluate both high-level planning and low-level control abilities of embodied agents. This benchmark comprises 100 complex everyday tasks based on scenarios from the Replica dataset, including long-horizon tasks, open-ended tasks with multiple possible outcomes, logical tasks, etc. To better assess agent performance, the authors propose three new metrics: Task Progress, Success End Rate, and Success Re-plan Rate. Additionally, the paper provides training data using Supervised Fine-Tuning (SFT) and Direct Preference Optimization (DPO), featuring Chain-of-Thought (CoT) outputs and replan processes. A baseline agent, HomieBot, is introduced, featuring a Large Multimodal Model (LMM) for high-level planning and several task-specific models for low-level control. HomieBot achieves a 31.8% success rate on training tasks and 20% on test tasks.

**Strengths:**

1. The objectives of the study are clearly stated.
2. The related work cited in this article is extensive.
3. The provision of the datasets (including CoT, replanning, etc.) is beneficial for future embodied AI research.

**Weaknesses:**

1. The article lacks clear organization.
     - In Section 3, it is inappropriate to describe the dataset for model training (Section 3.2, Data Augmentation for SFT and DPO) here. This dataset is intended for training the embodied agents, not for constructing the benchmark tasks.
     - The work aims to evaluate high-level planning and low-level execution abilities. However, there is no individual assessment of these abilities (e.g., the success rate of high-level planning, low-level execution, etc.). The relationship between the proposed evaluation metrics (TP, SER, SRR) and these abilities is unclear.
    - The problem is not well defined. It would be beneficial to provide a problem formulation regarding high-level planning and low-level execution in the paper.
   - Some content in the article is unclear. For instance, on page 4, line 207, the statement “if subtask i fails but subtask i + 1 succeeds” lacks clarity regarding the relationship between subtask i and subtask i+1. From my understanding, subtask i and i+1 are the same task but with different model responses (actions). This needs to be clarified.
2. There are a lot of mistakes in the paper,
   - In Section 3.3, Equation 1 appears to be incorrect. For different keypaths $k_i$, the nodes to be checked are also different. Therefore, the equation should be $TP=max_{{k_i}\in K_T} (\frac{len(k_i^{check})}{len(k_i)})$.
   - On page 6, line 278, M is defined as the set of all tasks, and $t \in M$, so $t$ should be a single task in the task space. However, the author also states “t is a task trajectory,” which conflates the concepts of task and trajectory.
   - The criterion “Success End Rate” is not reasonable. For example, if an agent never outputs “End” but can complete some tasks, the equation will be infinite. The correct equation should be $\frac{\text{number of successful tasks whose last output is End}}{\text{number of successful tasks}}$'.
   - On page 4, line 318, $i$ is not defined.
   - Spelling errors: On page 4, “controll” should be “control”.
3. The experimental section needs improvement. There is even no evaluation of the model before fine-tuning.

**Questions:**

See Weaknesses

---

> ### Author Response · Authors · 2024-11-23
> **Response to Weakness1**
>
> Thank you for your constructive comments. We have carefully considered your suggestions and would like to provide the following clarifications. We have also updated the manuscript, with the modified contents highlighted in orange for your review.
>
> ---
>
> **Q1: In Section 3, it is inappropriate to describe the dataset for model training (Section 3.2, Data Augmentation for SFT and DPO) here. This dataset is intended for training the embodied agents, not for constructing the benchmark tasks.**
>
> A1: Thank you for your suggestion on article organization, which is very helpful. We have moved the original Data Augmentation section to Section 4.1, the subsection before model training. We have also added more detailed of data augmentation in Appendix and supplementary materials.
>
> **Q2: The work aims to evaluate high-level planning and low-level execution abilities. However, there is no individual assessment of these abilities (e.g., the success rate of high-level planning, low-level execution, etc.). The relationship between the proposed evaluation metrics (TP, SER, SRR) and these abilities is unclear.**
>
> A2: We are sorry for the incomplete experiments and evaluation in the manuscript. Based on previous results, we refine the error definitions within Section 3.3, separating errors from each module to facilitate a comprehensive capability evaluation of HLP. By further classifying errors that are irrelevant to HLP, we can determine the success rate of each low-level skill. We conduct detailed error analysis in Section 4.5, all error statistics and low-level evaluation can be seen in Fig 4, Table 4 and Appendix H.2, we also provide more case studies in Appendix I. During the evaluation process, TP is primarily used to assess overall task completion, as mentioned in Section 2.2. It mainly reflects the evaluation of the task execution process and supplements the shortcomings of existing success rate evaluation methods. SER is closely related to the model's inherent judgment and logical reasoning abilities, while SRR mainly reflects the model's adjustment and generalization capabilities. We have added more analysis of the relationship between these metrics and the model's capabilities in Section 4.4.
>
> **Q3: The problem is not well defined. It would be beneficial to provide a problem formulation regarding high-level planning and low-level execution in the paper.**
>
> A3: Thank you for pointing out the issues in our paper writing and organization! We have added Section 3.1 to provide a detailed explanation of the EMMOE problem, as well as the functions of HLP and LLE. More specifically, the process of EMMOE tasks can be described as: the agent needs to make embodied decisions based on current environments and historical execution records in open environments, then navigates and manipulates within a continuous space, the obtained results and feedback will be used for the next decision. We divide this process into two main parts: HLP and LLE. HLP is responsible for embodied decision-making and planning adaptation while LLE handles continuous execution and provide feedback to HLP.
>
> **Q4: Some content in the article is unclear. For instance, on page 4, line 207, the statement “if subtask i fails but subtask i + 1 succeeds” lacks clarity regarding the relationship between subtask i and subtask i+1. From my understanding, subtask i and i+1 are the same task but with different model responses (actions). This needs to be clarified.**
>
> A4: We are sorry for the confusion caused by our unclear expression. As you inferred, for the $i$th subtask and its input instruction $I_i$, if the execution of model output $O_i$ fails but the next output $O_{i+1}$ succeeds after re-plan, we will choose $I_i$ as the *prompt*, $O_i$ as the *rejected* and $O_{i+1}$ as the *chosen*. We have also refined the algorithm description in Section 4.1 and provided the code implementation in Appendix F.2 and supplementary materials.

---

> > ### Author Response · Authors · 2024-11-23
> > **Response to Weakness2**
> >
> > **Q5: Incorrect TP equation; Conceptual confusion between *task* and *trajectory*.**
> >
> > A5: Thank you for your suggestion which would greatly improve the readability of the article. We revised the calculation equation of *TP* based on your advice and corrected the usage of *task* and *trajectory* in the article. Now *t* represents *trajectory* while *T* represents *task*, and *M* is defined as the set of all trajectories, not all tasks.
> >
> > **Q6: The criterion “Success End Rate” is not reasonable. For example, if an agent never outputs “End” but can complete some tasks, the equation will be infinite.**
> >
> > A6: Thank you for your questions about the metric calculation. We think this question is  caused by our unclear expression and conflation of *trajectory* and *task*. As we can the original equation of SER:
> > $$
> > \text{SER} = \frac{\text{len}(S)}{\sum_{t \in M} \text{count}_t(\text{end})}
> > $$
> > There are two factors to determine if a trajectory is successful: it must end with *End* and *TP* must reach 100%. When a trajectory meets the first factor, it will be counted into the denominator. But only when both two factors are satisfied will it be considered as a successful trajectory and counted into the numerator. Therefore, the case that *agent never outputs “End” but can complete some tasks* doesn’t exist, it will only be considered as a failed trajectory thus won’t be counted into SER calculation. The equation won't become infinite because the numerator will never exceed the denominator, which also means the upper limit of SER is 100%. We also add more calculation examples in Appendix B.1 for you to check.
> >
> > **Q7: On page 4, line 318, *i* is not defined; Spelling errors: On page 4, “controll” should be “control”.**
> >
> > A7: Thank you for your tolerance and suggestions on the writing. We carefully reviewed the article and addressed these problems. Additionally, the term *i*, which originally represents inventory, has been replaced with *inv* to avoid confusion.
> >
> > ---
> >
> > **Q8: The experimental section needs improvement. There is even no evaluation of the model before fine-tuning.**
> >
> > A8: We are sorry for the incomplete initial experiments. Now we have included evaluations of GPT-4o[1], Gemini-1.5-pro[2], Qwen2-VL-7B[3], MiniCPM-V 2.6[4]. The  results and comparisons are as follows:
> >
> > | Model                       | **SR**  | **PLWSR** | **TP**  | **SRR** | **SER**  |
> > |-----------------------------|---------|-----------|---------|---------|----------|
> > | GPT-4o     | 13.33   | 10.51     | 29.79   | 3.57    | 49.38    |
> > | Gemini-1.5-Pro      | 17.33   | 14.79     | 38.03   | 3.39    | 55.91    |
> > | Qwen2-VL-7B    | 1.00    | 0.50      | 16.55   | 0.59    | 25.00    |
> > | MiniCPM-V 2.6  | 0.67    | 0.57      | 14.45   | 0.06    | 40.00    |
> > | **HomieBot-7B (SFT)**                | 27.67   | 20.88     | 50.27   | **9.23** | 53.90    |
> > | **HomieBot-7B (SFT+DPO)**            | **30.30** | **24.66** | **51.39** | 8.72    | **60.81** |
> >
> > The analysis of the evaluation results can be referred to the content in Q2. More detailed of evaluation settings and model deployment can be found in Section 4.3 and Appendix H.1.
> >
> > ---
> >
> > **References**
> >
> > [1] GPT-4 Technical Report [arXiv:2303.08774](https://arxiv.org/abs/2303.08774)
> >
> > [2] Gemini 1.5: Unlocking multimodal understanding across millions of tokens of context [arXiv:2403.05530](https://arxiv.org/abs/2403.05530)
> >
> > [3] Qwen2-VL: Enhancing Vision-Language Model's Perception of the World at Any Resolution [arXiv:2409.12191](https://arxiv.org/abs/2409.12191)
> >
> > [4] MiniCPM-V: A GPT-4V Level MLLM on Your Phone [arXiv:2408.01800](https://arxiv.org/abs/2408.01800)

---

> > > ### Comment · Reviewer_tYVZ · 2024-11-26
> > > **Thanks the authors**
> > >
> > > I appreciate the authors’ responses and corrections, which have significantly improved the overall quality of the paper. However, I share the same opinion as Reviewer amMF: it would be beneficial for the authors to further refine and resubmit the paper.

---

> > > > ### Author Response · Authors · 2024-11-26
> > > >
> > > > Thank you for your response and for acknowledging our changes! We sincerely apologize for dificiencies in the initial submission. We also want to clarify that **our contributions remain unchanged**, which can be listed:
> > > >
> > > > * EMMOE, the first unified benchmark for both high-level and low-level embodied tasks with three novel metrics for more advanced evaluation.
> > > > * EMMOE-100, the first everyday task dataset featuring CoT outputs, diverse task design, re-plan processes, SFT and DPO dataset built on it.
> > > > * HOMIEBOT, a sophisticated agent system which integrates models at different levels, multiple error detection and adaptation mechanisms to complete EMMOE tasks.
> > > >
> > > > and **all methods used are the same as before**, everything from the previous version is included in the new version. **Our revisions focus mainly on new experimental designs and further analysis of the results, addressing the shortcomings mentioned by the reviewers**.
> > > > The additional materials include original code or data samples. I believe these updates are comprehensive, and we hope to make any possitive changes to our work based on your constructive suggestions, any criticism and feedback is acceptable.

---

> > > > > ### Author Response · Authors · 2024-11-26
> > > > > **Request for discussions**
> > > > >
> > > > > Would you mind spending a little more time to discuss the updated parts with us as the discussion period is extended? We would greatly appreciate your guidance and suggestions, as they would be extremely helpful to our work.

---

### Official Review · Reviewer_amMF · 2024-11-01

**Soundness:** 2
**Presentation:** 2
**Contribution:** 2
**Rating:** 5
**Confidence:** 4

**Summary:**

This paper presents a benchmark of 100 tasks in home robotics, collected within the Habitat simulator using human demonstrations annotated with detailed reasoning explanations. The benchmark evaluates models using success rate, success end rate, and success replan rate. The authors train Video-LLaVA on this dataset with either SFT (supervised fine-tuning) or DPO (Direct Preference Optimization) and find that SFT+DPO performs best on the train set across all metrics, while SFT-only achieves the highest scores on the test set.

**Strengths:**

- Introduction of a new benchmark for home robotics with a diverse set of 100 tasks.
- Collection of detailed annotations to capture the reasoning process and a diverse set of tasks in home environments.
- Strong metrics to evaluate embodied execution, focusing on nuanced aspects of embodied execution.
- Propose a dataset for SFT and DPO fine-tuning from egocentric trajectories.

**Weaknesses:**

- Limited baseline comparisons, which reduces the clarity of contributions relative to existing methods.
- Lack of diverse model comparisons, including text-only and zero-shot multi-image baselines (e.g., Qwen2VL, GPT-4o), and no modular vs. end-to-end performance analysis.
- Insufficient analysis of task-specific challenges and bottlenecks within the benchmark; error analysis and reasoning failure modes are explored to some degree, but not in detail.
- No discussion of potential real-to-sim discrepancies for the manipulation models, which could impact model generalization.

**Questions:**

- Could the authors provide stronger baseline comparisons, especially with text-only and zero-shot multi-image models?
- How do modular models compare to end-to-end approaches on this benchmark?
- Can the authors analyze the primary failures observed by the model, and give qualitative analysis of what are the major challenges in the benchmark for existing models?
- Is there any observed real-to-sim gap affecting the manipulation models, and if so, how does it impact performance?

---

> ### Author Response · Authors · 2024-11-23
> **Response to Weakness1-3**
>
> Thank you for your constructive comments. We have carefully considered your suggestions and would like to provide the following clarifications. We have also updated the manuscript, with the modified contents highlighted in orange for your review.
>
> ---
>
> **Q1: Limited baseline comparisons, which reduces the clarity of contributions relative to existing methods.**
>
> A1: Thank you for your beneficial suggestions! We have included evaluations of GPT-4o[1], Gemini-1.5-pro[2], Qwen2-VL-7B[3], MiniCPM-V 2.6[4] on the EMMOE dataset. The evaluation results and comparisons are as follows:
>
> | Model                       | **SR**  | **PLWSR** | **TP**  | **SRR** | **SER**  |
> |-----------------------------|---------|-----------|---------|---------|----------|
> | GPT-4o     | 13.33   | 10.51     | 29.79   | 3.57    | 49.38    |
> | Gemini-1.5-Pro      | 17.33   | 14.79     | 38.03   | 3.39    | 55.91    |
> | Qwen2-VL-7B    | 1.00    | 0.50      | 16.55   | 0.59    | 25.00    |
> | MiniCPM-V 2.6  | 0.67    | 0.57      | 14.45   | 0.06    | 40.00    |
> | **HomieBot-7B (SFT)**                | 27.67   | 20.88     | 50.27   | **9.23** | 53.90    |
> | **HomieBot-7B (SFT+DPO)**            | **30.30** | **24.66** | **51.39** | 8.72    | **60.81** |
>
> More detailed of evaluation settings and model deployment can be found in Section 4.3 and Appendix H.1.
>
> ---
>
> **Q2: Lack of diverse model comparisons, including text-only and zero-shot multi-image baselines (e.g., Qwen2VL, GPT-4o), and no modular vs. end-to-end performance analysis.**
>
> A2: Thank you for your valuable suggestions on our experiment. We have now included comparisons between different models and we would like to remind that EMMOE is a challenge including task planning, embodied decision-making, navigation and manipulation simultaneously. In the execution videos provided in our supplementary materials, you can see that our tasks are not only complex but also involve lengthy execution sequences. Current end-to-end models mainly focus on single visual navigation or tabletop manipulation and are not capable yet of completing all these tasks with a single model. Additionally, HomieBot and end-to-end models are not on the same level. HomieBot aims to build a platform that can evaluate different models simultaneously, which means end-to-end models can also possibly be evaluated in LLE module. The information transmitted from HLP to LLE is still relatively advanced natural language, and the subtask is more aligned with the "pick anything" or "go to anyplace" tasks studied in end-to-end models.
>
> ---
>
> **Q3: Insufficient analysis of task-specific challenges and bottlenecks within the benchmark; error analysis and reasoning failure modes are explored to some degree, but not in detail.**
>
> A3: Thank you for your insightful suggestions which have been extremely helpful in enhancing our work. First, we refined the error definitions within Section 3.3, allowing errors from different modules to be separated for easier statistical analysis. First, we further refined the error definitions in Section 3.3, separating errors across different modules to facilitate more straightforward statistical analysis and evaluation. Next, we collect all errors during the execution process and categorized them, examining the impact of each error on task performance. By further refining and analyzing these subcategories, we also determine the success rate of each low-level skill. Then we summarize several most predominant challenges and bottlenecks within the benchmark:
>
> * Physical grounding problems and model hallucinations.
>
> * Insufficient understanding ability when the execution steps become very long.
>
> * Limited low-level skill performance, especially *pick*.
>
> Section 4.5 contains a more detailed error analysis, all error statistics and low-level evaluation can be seen in Fig 4, Table 4 and Appendix H.2, we also conduct more case studies to further explore current bottlenecks in Appendix I.

---

> > ### Author Response · Authors · 2024-11-23
> > **Response to Weakness4**
> >
> > **Q4: No discussion of potential real-to-sim discrepancies for the manipulation models, which could impact model generalization.**
> >
> > A4: Your concerns about the real-to-sim gap are very sensible. In fact, we indeed observed several impacts on manipulation models, particularly in the following aspects:
> >
> > * Due to the limitations of the simulator, agents cannot receive the same feedback as in the real world. For example, during data collection and process execution, even minor collisions can result in unrealistic robotic arm states or cause it to pass through objects, which wouldn't happen in reality. These intrinsic limitations of simulator further reduce the model's ability.
> >
> > * The RT series[5, 6] has already pointed obvious generalization problems in visual language action(VLA) models. When the model meets tasks or scenery outside its training data, performance would drop drastically. This problem also occurs in real-to-sim gaps, and due to the embodiment’s ability in simulator can not match it in the real-world, the generalization issue is more pronounced. Now the low-level policies can be accurately evaluated on HomieBot are primarily trained within Habitat.
> >
> > ---
> >
> > **References**
> >
> > [1] GPT-4 Technical Report [arXiv:2303.08774](https://arxiv.org/abs/2303.08774)
> >
> > [2] Gemini 1.5: Unlocking multimodal understanding across millions of tokens of context [arXiv:2403.05530](https://arxiv.org/abs/2403.05530)
> >
> > [3] Qwen2-VL: Enhancing Vision-Language Model's Perception of the World at Any Resolution [arXiv:2409.12191](https://arxiv.org/abs/2409.12191)
> >
> > [4] MiniCPM-V: A GPT-4V Level MLLM on Your Phone [arXiv:2408.01800](https://arxiv.org/abs/2408.01800)
> >
> > [5] RT-1: Robotics Transformer for Real-World Control at Scale [arXiv:2212.06817](https://arxiv.org/abs/2212.06817)
> >
> > [6] RT-2: Vision-Language-Action Models Transfer Web Knowledge to Robotic Control [arXiv:2307.15818]( https://arxiv.org/abs/2307.15818)

---

> ### Comment · Reviewer_amMF · 2024-11-25
> **Thank you**
>
> I thank the authors for their response and the revisions made to the manuscript. The addition of new baselines and analysis has greatly strengthened the work, and I have raised my score to a 5. However, the significant changes to both the results and the manuscript make it challenging to fairly assess the current version in comparison to the original submission. For this reason, I still believe that a resubmission would be more appropriate.

---

> ### Author Response · Authors · 2024-11-25
>
> Thank you for your response and for acknowledging our changes! We sincerely apologize for dificiencies in the initial submission. We also want so clarify that **all methods in the new submission are entirely consistent with the previous version**, and all original experimental results are included in the new version. The **new additions aim to address the reviewers' concerns and enhance the work**, and **our contributions stay unchanged**. We hope to make any possitive changes to our work based on your constructive suggestions!

---

> > ### Author Response · Authors · 2024-11-26
> > **Request for discussion**
> >
> > Would you mind spending a little more time to discuss the updated parts with us due to the extended discussion period? We would greatly appreciate your guidance and suggestions, as they would be extremely helpful to our work.

---

### Official Review · Reviewer_VZgA · 2024-11-03

**Soundness:** 2
**Presentation:** 1
**Contribution:** 1
**Rating:** 3
**Confidence:** 4

**Summary:**

This work focuses on studying embodied AI instruction following mobile manipulation in simulated environments. To this end, the paper proposes the EMMOE-100 dataset and benchmark and HOMIEBOT as a system integration solution to it. Some unique contributions involve that it is a dataset of human-controlled embodied agent trajectories:
1. Decomposition into subtasks
1. Annotated reasoning process with every execution
1. Replanning for some failed subtasks
The paper also introduces metrics to measure Task Progress, Success end rate, and success replan rate and presents results for SFT and SFT+DPO on the Video-LlaVA model.

**Strengths:**

1. The paper introduces the human controlled embodied agent trajectories, unlike the use of PDDL in ALFRED and other RL benchmarks.
1. The focus seems to be on open-ended long horizon questions, multiple ways of solving it, annotating sub-goals and deliberately collecting replanning trajectories.

**Weaknesses:**

1. Overclaim: it does not seem like the first to unify the high-level and low-level. This has been discussed in Robotics Task and Motion planning (TAMP) literature and has been introduced as baselines in previous simulator benchmarks. Connection with previous work and clarity on the unique contribution of this work is needed.
1. Lack of failure analysis: The paper does not discuss the limitations and failures modes of the combined high-level and low-level modules.
1. Limited discussion: It is unclear why the success rate for HOMIEBOT is so low in training and test tasks.
1. Clarity: Paper is not well formatted and has lots of typos (line 180 control, 185-186 between)
1. Limited novelty: with discrete action space, the work looks very similar to ALFRED (Shridhar et al.) and FiLM (Min et al.) with a few enhancements.

**Questions:**

1. Why not use continuous action space? Habitat Replica allows for continuous action space. HM3D
2. How is the Success end rate, and success re-plan rate measured? How is the count for end and replan computed for each trajectory?
3. Why is “trainable data format” listed as an important contribution compared to the existing datasets?
4. How does the high-level planner know when and where low-level execution fails?

---

> ### Author Response · Authors · 2024-11-23
> **Response to Weakness1-5**
>
> Thank you for your constructive comments. We have carefully considered your suggestions and would like to provide the following clarifications. We have also updated the manuscript, with the modified contents highlighted in orange for your review.
>
> ---
>
> **Q1: Overclaim: it does not seem like the first to unify the high-level and low-level. This has been discussed in Robotics Task and Motion planning (TAMP) literature and has been introduced as baselines in previous simulator benchmarks. Connection with previous work and clarity on the unique contribution of this work is needed.**
>
> Thank you for your suggestion! We think this misunderstanding stems from our unclear definitions of HLP and LLE in the article. The process of EMMOE tasks can be described as: the agent needs to make embodied decisions based on current environments and historical execution records in open environments, then navigates and manipulates within a continuous space, the obtained results and feedback will be used for the next decision. We divide this process into two main parts: High Level Planning (HLP) and Low Level Execution (LLE). HLP is responsible for embodied decision-making and planning adaptation while LLE handles continuous execution and provide feedback to HLP. From these definitions, it is clear that our HLP differs from the high-level in TAMP. Making decisions in open environments means we won't provide any background information to the planner, it must rely on its own capabilities to make decisions and dynamically interact with the environment. Current embodied decision-making often focuses solely on planning, simplifying the execution to varying degrees. However, a more reasonable evaluation should assess both high-level and low-level processes together, as execution and decision-making are interconnected, with execution influencing decision-making. This is also one of the motivations behind the proposal of EMMOE.
>
> ---
>
> **Q2: Lack of failure analysis: The paper does not discuss the limitations and failures modes of the combined high-level and low-level modules.**
>
> We are sorry for the incomplete discussions of our methods. We refine the error definitions within Section 3.3, separating errors from each module to explore limitations and bottlenecks in current modules. By further classifying errors that are irrelevant to HLP, we determine the success rate of each low-level skill. We conduct detailed error analysis in Section 4.5, all error statistics and low-level evaluation can be seen in Fig 4, Table 4 and Appendix H.2, we also provide more case studies in Appendix I.
>
> ---
>
> **Q3: Limited discussion: It is unclear why the success rate for HOMIEBOT is so low in training and test tasks.**
>
> We thank you for your attention to our experimental results. We add more comparisons with other LMMs, combining methods and conclusions mentioned in Q2, we attribute it to the following reasons:
>
> * Insufficient understanding ability when the execution steps become very long.
>
> * Physical grounding problems and model hallucinations.
>
> * Limited low-level skill performance, especially *pick*.
>
> * The phenomenon of historical forgetting during model inference.
>
> Notably, we measure the average success rate of trajectories, not how many tasks can be completed. This method of measurement is certainly lower than the latter but is more stable and fair. Detailed analysis can be found in Section 4.4 and Section 4.5.
>
> ---
>
> **Q4: Clarity: Paper is not well formatted and has lots of typos (line 180 control, 185-186 between)**
>
> A4: We are sorry for the unclear writing and we have reorganized the manuscript and corrected the grammatical errors.
>
> ---
>
> **Q5: Limited novelty: with discrete action space, the work looks very similar to ALFRED (Shridhar et al.) and FiLM (Min et al.) with a few enhancements. Why not use continuous action space? Habitat Replica allows for continuous action space. HM3D**
>
> A5: We think this misunderstanding was raised by our wrong description. Actions for the HLP are discrete and require selection from an available action list. However, once an action is selected, it will be executed in continuous space. The agent must perform each task in continuous space and receives feedback from the actual interaction, then send them to HLP. This is also what previous work based on the ALFRED dataset couldn't achieve. Due to the limitations of the simulator, the agent couldn't perform actions in continuous space or obtain real interaction information. We have revised the inappropriate description of the dataset in the manuscript and we strongly recommend viewing the demonstration video provided in the supplementary materials, where can clearly see that the actions are executed in continuous space.

---

> > ### Author Response · Authors · 2024-11-23
> > **Response to Question1-4**
> >
> > **Q6: How is the Success end rate, and success re-plan rate measured? How is the count for end and replan computed for each trajectory?**
> >
> > Each execution trajectory will be saved with the historical dialogue, images and video, from which we can extract all the execution information, including the execution's termination and the number of replan. The dialogue format can be found in Appendix F, I and supplementary materials. We also provide a more detailed calculation process for SER and SRR in Appendix C.
> >
> > ---
> >
> > **Q7: Why is “trainable data format” listed as an important contribution compared to the existing datasets?**
> >
> > Thank you for your attention to our work's contributions. As the challenges mentioned in page 2, lines 63-72, there are several reasons for the significance of the trainable data format.
> >
> > * In mobile manipulation, data scarcity and the resulting task generalization are common challenges. By splitting long trajectories into smaller steps, lowe-level models can specialize in specific actions, while high-level planners handles the coordination of these skills. This methods not only reduces data demands but also can harness LMMs' reasoning power to complete tasks beyond the original data.
> >
> > * Robotics data for IL or RL is always not trainable for LLMs, which require dialogue style data. LLM prefer to output human-style instructions, whereas action execution requires more precise and practical instructions. With a LMM-trainable data, LMM can quickly build up a general understanding of the current environment, enabling outputs to be grounded and compatible with low-level models.
> >
> > The error analysis in Section 4.5 effectively supports our views: without training, LMMs will struggle to achieve physical grounding, resulting in model hallucinations and a large number of non-executable outputs. This issue has been significantly improved in our models, which also highlights the significance of LMM-trainable format data.
> >
> > ---
> >
> > **Q8: How does the high-level planner know when and where low-level execution fails?**
> >
> > Thank you for your attention to our methods. Error detection will occur at different stages during the whole low-level execution. Before execution, we perform an initial error detection based on current inventory, target information and the following action. Execution proceeds only if no errors are detected. During execution, feedback from low-level models is monitored, if execution fails, the details are logged. After execution, a final error detection is conducted using the execution details, current state of the robotic arm and the updated inventory. Then the feedback will be sent to high-level planner. We provide a detection pipeline in Appendix E.1 for you to check.

---

> > > ### Comment · Reviewer_VZgA · 2024-11-26
> > > **Response to the Rebuttal**
> > >
> > > I appreciate the efforts to make significant clarifications and updates to the paper, but it makes difficult to thoroughly review everything. I have decided to maintain my original score as of now.
> > >
> > > Two questions:
> > >
> > > 1. Limited error detection: I understand the different categories described in Sec 3 and Appendix E. However, I think this approach for rule-based error detection is hard to scale for general-purpose. For example, you say,
> > > ```
> > > if action == ’put’ and "closed" in check_status(target):
> > >     return ’fail’, f’Unable to put, the {target} is closed, you should open it first’
> > > ```
> > > My concern: you do not handle the scenarios for put something "in a sink", "on a shelf", etc.
> > >
> > > 2. Elaborate discussion on why this is different from other solutions in the area of TAMP: "From these definitions, it is clear that our HLP differs from the high-level in TAMP." - I disagree. Kindly refer to [1, 2]. The main claim of providing feedback to HLP with error detection seems like a symbolic approach - written as if-else instead of PDDL or satisfaction constraints.
> > >
> > > [1] Integrated Task and Motion Planning, by Garrett et al. (2020)
> > >
> > > [2] Combining task and motion planning: Challenges and Guidelines, by Mansouri et al. (2021)

---

> > > > ### Author Response · Authors · 2024-11-26
> > > >
> > > > Thank you for your response and suggestions!
> > > >
> > > > **Q1: Limited error detection: I understand the different categories described in Sec 3 and Appendix E. However, I think this approach for rule-based error detection is hard to scale for general-purpose.**
> > > >
> > > > First, it should be clarified that the pipeline code clip in Appendix is a modified version of the original code (the original code is too lengthy to put into the article).
> > > > The *check_status* function is a modified generalized interface which implementation depends on different simulators.
> > > >
> > > > Second, the case mentioned can actually be detected. The purpose of this function is to check whether any of the object's properties include "closed." For objects like *table* or *sink*, they have no *open* or *closed* properties, so the function won't return *closed*. Therefore, the action *put into sink* is entirely achievable. This can also be observed in the examples provided in our appendix, which include similar outputs.
> > > >
> > > > **Q2: Elaborate discussion on why this is different from other solutions in the area of TAMP: "From these definitions, it is clear that our HLP differs from the high-level in TAMP." - I disagree. Kindly refer to [1, 2]. The main claim of providing feedback to HLP with error detection seems like a symbolic approach - written as if-else instead of PDDL or satisfaction constraints.**
> > > >
> > > > Here we want to reclaim the precondition of HLP: HLP is responsible for embodied decision-making and planning adaptation while LLE handles continuous execution and provide feedback to HLP. Making decisions in open environments means **no background information and specific target information will be provided and the worlds are unknown**, it must rely on its own capabilities to make decisions and dynamically interact with the environment.
> > > >
> > > > The **inputs and outputs of HLP are all natural language rather than symbolic language**, HLP leans more toward the integration of embodied decision-making problems defined in [1] and real execution feedback adaptation, rather than the traditional high-level planning in TAMP.
> > > >
> > > > **References**
> > > >
> > > > [1] Embodied Agent Interface: Benchmarking LLMs for Embodied Decision Making  [arXiv:2410.07166](https://arxiv.org/abs/2410.07166)

---

### Official Review · Reviewer_piLp · 2024-11-04

**Soundness:** 2
**Presentation:** 2
**Contribution:** 2
**Rating:** 5
**Confidence:** 3

**Summary:**

The paper introduces three main contributions:

1. EMMOE: A unified benchmark for evaluating both high-level planners and low-level policies in embodied mobile manipulation tasks.

2. EMMOE-100: A dataset with 100 complex everyday tasks featuring task-planning processes and COT outputs

3. HOMIEBOT: An intergrated agent system combining high-level planner and low-level executors.

**Strengths:**

1. The new evaluation metrics are novel, potentially providing a valuable resource for the research community and practical applications.

**Weaknesses:**

1. **Insufficient Methodology Details**
   - The data processing and augmentation methods are not adequately explained:
     * The "fixed-format conversation data" conversion process in SFT Augmentation is not described
     * No clear explanation of how the uniform script processes the EMMOE-100 data
     * The DPO Augmentation section lacks clear algorithmic description or flowcharts

2. **Limited Experimental Evaluation**
   - Inadequate baseline comparisons and empirical analysis:
     * No evaluation of existing RL algorithms or LM-based agents on the EMMOE benchmark
     * Only one system (HOMIEBOT) is evaluated, lacking comparative analysis
     * The relatively low success rates (31.8% training, 20% testing) are not thoroughly analyzed
     * Missing ablation studies or detailed error analysis to understand performance bottlenecks

**Questions:**

See weaknesses.

---

> ### Author Response · Authors · 2024-11-23
> **Response to Weakness 1-2**
>
> Thank you for your constructive comments. We have carefully considered your suggestions and would like to provide the following clarifications. We have also updated the manuscript, with the modified contents highlighted in orange for your review.
>
> ---
>
> **Q1: Insufficient Methodology Details. The data processing and augmentation methods are not adequately explained**
>
> A1: Thank you for your helpful suggestions and attention to the details of our methods. We have revised the description of the DPO Augmentation in Section 4.1 to make it more understandable and clearer. Additionally, we've added the code implementations for SFT and DPO Augmentation in Appendix F, along with samples of the converted SFT and DPO data. These codes will convert the EMMOE data shown in Appendix B into a dialogue format suitable for training. Furthermore, we've released the anonymized code for data augmentation in our supplementary materials to facilitate understanding.
>
> ---
>
> **Q2: No evaluation of existing RL algorithms or LM-based agents on the EMMOE benchmark**
>
> A2: Thank you for pointing out the shortcomings in our work! We have added evaluations of GPT-4o[1], Gemini-1.5-pro[2], Qwen2-VL-7B[3], MiniCPM-V 2.6[4] on the EMMOE dataset. The evaluation results are shown as follows:
>
> | Model                       | **SR**  | **PLWSR** | **TP**  | **SRR** | **SER**  |
> |-----------------------------|---------|-----------|---------|---------|----------|
> | GPT-4o     | 13.33   | 10.51     | 29.79   | 3.57    | 49.38    |
> | Gemini-1.5-Pro      | 17.33   | 14.79     | 38.03   | 3.39    | 55.91    |
> | Qwen2-VL-7B    | 1.00    | 0.50      | 16.55   | 0.59    | 25.00    |
> | MiniCPM-V 2.6  | 0.67    | 0.57      | 14.45   | 0.06    | 40.00    |
> | **HomieBot-7B (SFT)**                | 27.67   | 20.88     | 50.27   | **9.23** | 53.90    |
> | **HomieBot-7B (SFT+DPO)**            | **30.30** | **24.66** | **51.39** | 8.72    | **60.81** |
>
> More detailed of evaluation settings and model deployment can be found in Section 4.3 and Appendix H.1.
>
> ---
>
> **Q3: Only one system (HOMIEBOT) is evaluated, lacking comparative analysis**
>
> A3: Thank you for your valuable suggestions on our work! First, we would like to clarify that EMMOE is a new challenge and differs from previous benchmarks. Previous benchmarks often simplify either the decision-making or practical execution aspects to evaluate a single capability, while EMMOE simultaneously addresses multiple challenges, including task planning, embodied decision-making, navigation and manipulation. Due to the differences in tasks, existing systems are not fully applicable to EMMOE. To our knowledge, HomieBot is currently the only system capable of performing all these evaluations simultaneously. Secondly, the main contribution of this work lies in proposing a new benchmark and comprehensive evaluation methods. In the newly added Section 4.5, we demonstrate how HomieBot can be used to evaluate different models. HomieBot serves more like a platform where various models or systems can be integrated after adjustments, as shown in Appendix H.1.
>
> ---
>
> **Q4: The relatively low success rates (31.8% training, 20% testing) are not thoroughly analyzed. Missing ablation studies or detailed error analysis to understand performance bottlenecks**
>
> A4: Thank you for your instrumental suggestions in improving our work. Here's how we addressed them and make performance analysis:
>
> * We refine the error definitions in Section 3.3, enabling errors from different modules to be separated for more straightforward statistical analysis and evaluation.
>
> * We collect all errors occurred during evaluation process and make a detailed classification, evaluating the impact of each error on task performance. By further analyzing these subcategories, we determine the success rate of each low-level skill.
>
> * We conduct more case studies to further explore the current bottlenecks in Appendix I.
>
> Our main findings of current bottlenecks hindering the model’s performance can be summarize as:
>
> * Issues with physical grounding and model hallucinations.
>
> * Insufficient understanding capacity when execution steps become excessively long.
>
> * Limited performance of low-level skills, particularly in action *pick*.
>
> Section 4.5 provides a more detailed error analysis, with comprehensive error statistics and low-level evaluations presented in Figure 4, Table 4, and Appendix H.2.
>
> ---
>
> **References**
>
> [1] GPT-4 Technical Report  [arXiv:2303.08774](https://arxiv.org/abs/2303.08774)
>
> [2] Gemini 1.5: Unlocking multimodal understanding across millions of tokens of context [arXiv:2403.05530](https://arxiv.org/abs/2403.05530)
>
> [3] Qwen2-VL: Enhancing Vision-Language Model's Perception of the World at Any Resolution [arXiv:2409.12191](https://arxiv.org/abs/2409.12191)
>
> [4] MiniCPM-V: A GPT-4V Level MLLM on Your Phone [arXiv:2408.01800](https://arxiv.org/abs/2408.01800)

---

> > ### Comment · Reviewer_piLp · 2024-11-26
> >
> > Thank the authors for providing new results, code, and examples to better explain their method. The paper becomes clearer and more convincing. I have raised my overall rating to 5. However, there is so much new content added not only in the results but also in the methodology. It is challenging to make a fair assessment.

---

> > > ### Author Response · Authors · 2024-11-26
> > >
> > > We are so thankful for your kindness and agreements on our changes! We can ensure that the methodology part is same as before, even the most different error detection part is refined from it, as you can see in the supplimentaries and appendix, or compare with the revision history. We thank again for your constructive advice and wait further discussions.

---

### Author Response · Authors · 2024-11-23

We thank all reviewers for your interest in and suggestions for our work. We are also sorry for the shortcomings in our manuscript due to time constraints and lack of experience, which may have caused inconvenience to you, we are grateful for your tolerance and guidance, which have helped us complete this manuscript. In the updated version, we primarily make the following changes based on the reviewers' feedback and mark them in orange:

* **Paper Writing and Organization:** We move the Data Augmentation part to Section 4.1, add Section 3.1 for more precise system definitions, rewrite Section 4.4 to emphasize the relationship between new metrics and results, add Section 4.5 for detailed performance analysis. We also correct writing errors, remove redundant descriptions, and move related work to Appendix A due to the page limitation.

* **Methodology Details:** We refine the method descriptions and correct mistakes mentioned by reviewers for clarity and accuracy. Additionally, we add more detailed task demonstrations, metric calculations, code implementations for data augmentation, data samples, visualization processes for system and experimental details in Appendix B~H.

* **Comparison Experiments:** We add evaluations of GPT-4o[1], Gemini-1.5-pro[2], Qwen2-VL-7B[3], MiniCPM-V 2.6[4] on the EMMOE dataset, evaluation settings and model deployment can be found in Section 4.3 and Appendix H.1. The analysis of the results can be found in Section 4.4.

* **Performance Analysis:** We refine the error definitions in Section 4.5 and conduct comprehensive error statistics to further explore bottlenecks in the current challenge, we also show how to use HomieBot to evaluate both high-level planners and low-level policies, the results are in Figure 4, Table 4, and Appendix H.2. We also conduct several case studies in Appendix I.

* **Supplimentary Materials:** We add more anonymized codes, scripts and data demonstrations in our supplimentary materials. After further organization, all codes and data will be open-sourced to make contributions to the community and ensure the reproducibility.

At the same time, we would also like to clarify that:

**Our contributions remain unchanged**, which can be listed:

* EMMOE, the first unified benchmark for both high-level and low-level embodied tasks with three novel metrics for more advanced evaluation.
* EMMOE-100, the first everyday task dataset featuring CoT outputs, diverse task design, re-plan processes, SFT and DPO dataset built on it.
* HOMIEBOT, a sophisticated agent system which integrates models at different levels, multiple error detection and adaptation mechanisms to complete EMMOE tasks.

Our **core contributions are the new benchmark, metrics and evaluation methods**, followed by the design of HomieBot. Our primary requirement for HomieBot is to robustly perform EMMOE tasks. In revisions, **all methods and data used are the same as before**(including EMMOE-100, SFT and DPO augmentation, HLP and LLE frame), everything from the previous version is included in the new version. **Our revisions focus mainly on experimental comparisons and further analysis of the results, addressing the shortcomings mentioned by the reviewers and highlighting our features**.
The additional materials include plenty of original code or data samples. I believe these updates are comprehensive, and we hope to engage in further discussion with you. We welcome any criticism and feedback!

---

### Author Response · Authors · 2024-12-01

We thank all the reviewers once again for your valuable suggestions and the time and effort dedicated to our work. As the discussion phase comes to a close, we hope to receive further feedback on the article's content and engage in in-depth discussions with you. We are committed to continuously polishing our work and answering all questions you may have, and we will respond as quickly as possible after receiving feedback!

---

### Note · Authors · 2025-01-22

I have read and agree with the venue's withdrawal policy on behalf of myself and my co-authors.